# Unconventional secretory processing diversifies neuronal ion channel properties

Cyril Hanus[1]*, Helene Geptin[1], Georgi Tushev[1], Sakshi Garg[1], Beatriz Alvarez-Castelao[1], Sivakumar Sambandan[1], Lisa Kochen[1], Anne-Sophie Hafner[1], Julian D Langer[1,2], Erin M Schuman[1]*

[1]Max Planck Institute for Brain Research, Frankfurt, Germany; [2]Max Planck Institute for Biophysics, Frankfurt, Germany

*For correspondence: cyril. hanus@brain.mpg.de (CH); erin. schuman@brain.mpg.de (EMS)

**Competing interests:** The authors declare that no competing interests exist.

**Abstract** N-glycosylation – the sequential addition of complex sugars to adhesion proteins, neurotransmitter receptors, ion channels and secreted trophic factors as they progress through the endoplasmic reticulum and the Golgi apparatus – is one of the most frequent protein modifications. In mammals, most organ-specific N-glycosylation events occur in the brain. Yet, little is known about the nature, function and regulation of N-glycosylation in neurons. Using imaging, quantitative immunoblotting and mass spectrometry, we show that hundreds of neuronal surface membrane proteins are core-glycosylated, resulting in the neuronal membrane displaying surprisingly high levels of glycosylation profiles that are classically associated with immature intracellular proteins. We report that while N-glycosylation is generally required for dendritic development and glutamate receptor surface expression, core-glycosylated proteins are sufficient to sustain these processes, and are thus functional. This atypical glycosylation of surface neuronal proteins can be attributed to a bypass or a hypo-function of the Golgi apparatus. Core-glycosylation is regulated by synaptic activity, modulates synaptic signaling and accelerates the turnover of GluA2-containing glutamate receptors, revealing a novel mechanism that controls the composition and sensing properties of the neuronal membrane.

## Introduction

Most membrane and secreted proteins are N-glycosylated during their synthesis and processing in the secretory pathway (*Moremen et al., 2012*). During this process, as nascent proteins emerge in the lumen of the endoplasmic reticulum (ER), a mannose-rich precursor is first transferred *en bloc* to specific asparagine residues. These immature 'core-glycans' are then trimmed down and modified by the sequential addition of diverse monosaccharides as proteins exit the ER and progress through the Golgi apparatus before they are sent to their final destination (*Moremen et al., 2012*; *Aebi et al., 2010*). This sequential and combinatorial modification results in a huge potential diversity of N-glycans and regulates virtually every aspect of membrane protein biology, in particular protein folding, trafficking, stability, ligand-binding and interaction with the extracellular matrix (*Moremen et al., 2012*; *Aebi et al., 2010*; *Scott and Panin, 2014*; *Miller and Aricescu, 2014*). Consequently, congenital N-glycosylation defects, especially in the brain, result in severe and often lethal developmental disorders (*Cylwik et al., 2013*). Although the primary organelles of the secretory pathway were first described in neurons (*Golgi, 1989*; *Nissl, 1903*), little is known about the N-glycosylation of neuronal membrane proteins.

Numerous mRNAs encoding surface and secreted proteins are localized to dendrites (*Cajigas et al., 2012*). Yet, how dendritic secretory cargo is processed after local synthesis is still

**eLife digest** Information is carried around the nervous system by cells called neurons. The ability of neurons to communicate with each other relies on many proteins that are found on the surfaces of the cells. Like in all animal cells, surface proteins are made inside the cell in a compartment called the endoplasmic reticulum. During this process, one or several complex sugar molecules are usually added to newly made proteins. These sugar molecules are then modified as the proteins leave the endoplasmic reticulum and pass through another compartment called the Golgi apparatus on the way to the cell membrane. The precise number and structure of the sugar molecules attached to the protein define its glycosylation profile.

Neurons receive information from other neurons at branch-like structures called dendrites, which trigger electrical signals that travel through the rest of the cell. To directly control how dendrites generate these signals, neurons make surface proteins locally in dendrites. However, while the endoplasmic reticulum is found all over the neuron, including in the dendrites, the Golgi apparatus is generally only present in the main cell body. It is not known how surface proteins are made in the dendrites or how the proteins' glycosylation profiles are altered in the absence of a Golgi apparatus.

Hanus et al. used microscopy and biochemical techniques to study the glycosylation profiles of surface proteins in rat neurons. The experiments revealed that immature glycosylation profiles are found on hundreds of different proteins that have been transported to the cell surface. This includes many proteins that are needed to transmit electrical signals between neurons. Next, Hanus et al. selectively blocked the modification of sugar molecules on proteins in the Golgi apparatus. This showed that dendrites are able to form and work properly even if surface proteins have primarily immature glycosylation profiles.

Further experiments suggest that immaturely glycosylated proteins might travel to the surface of neurons using a different route that bypasses the Golgi apparatus. The next step will be to investigate exactly how these proteins are delivered to the surface and how they influence the way neurons behave.

debated. In mammals, neuronal dendrites contain ER, ER-exit sites and ER-Golgi intermediate compartments (ERGIC) and occasionally Golgi outposts, but, for the most part lack canonical Golgi membranes (*Torre and Steward, 1996*; *Gardiol et al., 1999*; *Krijnse-Locker et al., 1995*; *Horton and Ehlers, 2003*; *Hanus and Ehlers, 2008*; *Cui-Wang et al., 2012*; *Hanus et al., 2014*). It is thus conceivable that, depending on their synthesis in the soma or dendrites, nascent cargo visits distinct sets of secretory subcompartments, and hence acquire specific types of N-glycans. For example, it is believed that nascent neurotransmitter receptors may follow multiple and specific secretory itineraries (*Jeyifous et al., 2009*), but whether this impacts their glycosylation is unknown.

Intriguingly, Concanavalin A (ConA), a mannose-binding lectin, has been widely used to block the desensitization of plasma-membrane localized AMPA and kainate glutamate receptors (*Reiner and Isacoff, 2014*; *Hoffman et al., 1998*; *Evans and Usherwood, 1985*). From a cell-biological perspective, this is puzzling as core-glycosylated (mannose-rich) N-glycans are typically not found at the cell-surface (*Moremen et al., 2012*; *Aebi et al., 2010*; *Grieve and Rabouille, 2011*). Indeed, a specific sensitivity to the glycosidase EndoH (which cleaves mannose-rich glycans) is commonly used to identify intracellular immature membrane proteins in total cellular lysates (*Shi et al., 2010*; *Greger et al., 2002*; *Tomita et al., 2003*; *Sans et al., 2001*; *Tucholski et al., 2013a*).

Here we demonstrate that hundreds of neuronal surface membrane proteins are indeed core-glycosylated, resulting in the neuronal membrane displaying atypically high and activity-dependent levels of ConA-reactive species. We found that while N-glycosylation is generally required for the proper expression of membrane proteins at the neuronal surface, 'immature' core-glycosylated proteins are sufficient to sustain dendritic development and synaptic transmission, indicating that these proteins are fully functional. Focusing on candidate neurotransmitter receptors and auxiliary subunits, we show that core-glycosylated proteins access the cell-surface in a Golgi-independent manner indicating a bypass or a hypo-function of the Golgi apparatus. This atypical N-glycosylation results in an accelerated turnover of membrane proteins and modulates synaptic signaling, revealing a novel

mechanism controlling membrane protein homeostasis and function in morphologically complex cells such as neurons.

## Results

### Core-glycosylated proteins are atypically abundant at the neuronal surface

Their specific binding and sensitivity to distinct lectins and glycosidases distinguishes 3 basic types of N-glycans on membrane proteins: core-glycosylated (or 'immature'), hybrid and complex (*Figure 1A*; and see Materials and methods for details) (*Moremen et al., 2012*; *Scott and Panin, 2014*; *Zielinska et al., 2010*). To compare the surface levels of these three generic N-glycan types in neurons to non-neuronal cells, we labeled mixed (neuron-glia) hippocampal cultures and 3 commonly used cell-lines (COS7, BHK and CHO) with different lectin biotin-conjugates under non-permeabilizing conditions. As expected, all cell types examined (including neurons and glia) displayed both hybrid and complex N-glycans (labeled by RCA and WGA, respectively) (*Figure 1B*). Surprisingly, however, mature neurons (*Figure 1—figure supplement 1*) also displayed a high level of core-glycosylated N-glycans (strongly labeled by ConA and GNA) (*Figure 1B*). This observation was reinforced by the comparison of neurons and several other cell types: only neurons showed a prominent surface labeling by ConA and GNA, confirming high levels of core-glycosylated proteins at the neuronal plasma membrane (*Figure 1C*).

To confirm the specificity of the labeling, neurons were labeled with the same lectins after treatment with Kifunensine (Kf), an inhibitor of ER and Golgi type I mannosidases, which prevents the maturation of core-glycans into hybrid and complex glycans (*Tulsiani et al., 1982*). As expected, Kf increased the surface levels of ConA and GNA-reactive glycans and reduced the surface levels of RCA and WGA-reactive glycans (*Figure 1—figure supplement 2*), thus confirming the presence of core-glycosylated proteins on the neuronal surface. To verify that intracellular proteins were indeed localized in the expected secretory compartment, we labeled neurons with ConA, RCA and WGA under permeabilizing conditions together with anti-MAP2 antibodies to visualize the somatodendritic compartment. As expected, ConA marked ER-like structures (*Cui-Wang et al., 2012*) while RCA and WGA labeled distinct Golgi-like membranes and, for WGA, the nuclear envelope (EN) (*Figure 1—figure supplement 3*). We also validated the labeling of these intracellular compartments in COS7 cells, where the clear morphology of secretory organelles allows one to unambiguously identify various intracellular structures, and obtained similar results (*Figure 1—figure supplement 4*). Altogether, these data thus indicate that the atypical abundance of ConA-reactive species at the neuronal surface is indeed due to core-glycosylated proteins.

How abundant is the core-glycosylation of neuronal membrane proteins? To compare directly the relative expression of the different glycans at the neuronal cell surface, we fractioned surface and intracellular proteins by affinity purification after surface biotinylation (*Figure 2A* and *Figure 2—figure supplement 1*) and quantified the levels of ConA, RCA and WGA reactive-glycans in these two fractions by far-western blotting (see Materials and methods). We again found, both in cultured neurons and brain tissue, that core-glycans (ConA-reactive species) were surprisingly abundant at the neuronal surface (*Figure 2B*). Control experiments in total neuronal extracts treated with PNGase and EndoH showed that only ConA-reactive species were sensitive to EndoH, thus confirming that these were core-glycosylated N-glycans (*Figure 2—figure supplement 2*). Interestingly, quantification of the relative surface expression of ConA, RCA and WGA reactive species showed that, in neurons, core-glycosylated proteins were expressed at relative levels comparable to conventional 'mature' N-glycans (*Figure 2C*).

### A number of surface neurotransmitter receptors display atypical N-glycosylation profiles

Where are core-glycosylated proteins localized in the neuronal plasma membrane? To address this, we imunolabeled synapses with the presynaptic marker protein bassoon after surface labeling with ConA. Core N-glycans were distributed throughout the entire neuronal surface but were particularly abundant at synapses, notably in dendritic spines (*Figure 3A*). To examine whether synaptic proteins themselves displayed core-glycosylation profiles, we assessed the glycosylation status of a cast of

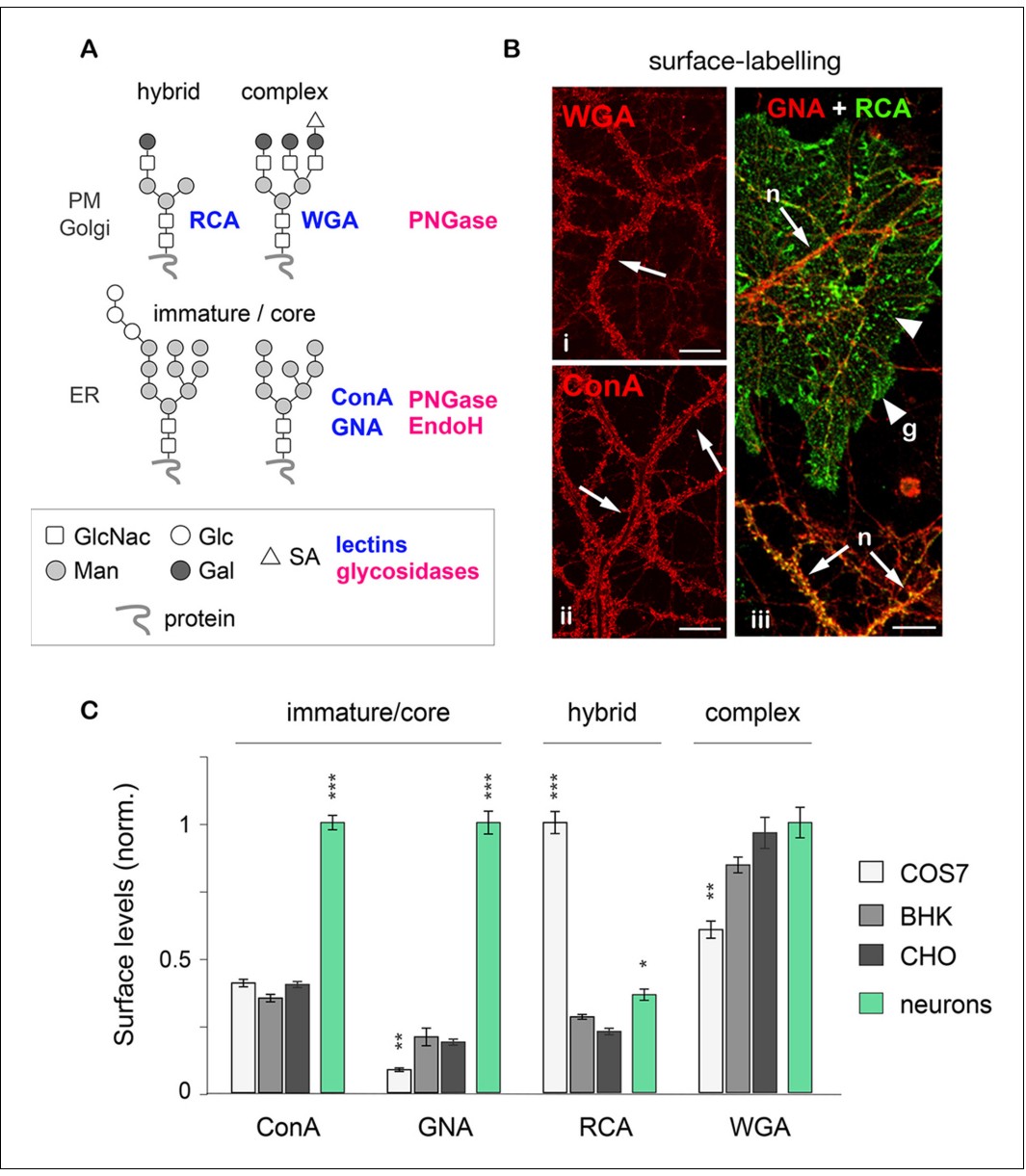

**Figure 1.** Core-glycosylated proteins are abundant at the neuronal surface. (**A**) The binding to specific lectins and the sensitivity to distinct glycosydases distinguish different N-glycans. Hybrid and complex N-glycans detected at the plasma membrane (PM) and the Golgi apparatus (Golgi) bind ricin agglutinin (RCA) and wheat germ agglutinin (WGA), respectively, and are hydrolyzed by PNGase. In contrast, core-glycosylated N-glycans are usually specifically localized to the endoplasmic reticulum (ER), bind concanavalin A and galanthus nivalis agglutinin (GNA) and are hydrolyzed by both PNGase and EndoH. (**B**) Cultured hippocampal neurons (DIV 40) after surface labeling with ConA, GNA, WGA (red) and RCA (green) to identify core-glycosylated (ConA, GNA), complex (WGA) and hybrid (RCA) glycans, respectively. In iii, 'n' and 'g' mark neurons and glial cells, respectively. Note the high reactivity of the neuronal surface to ConA, WGA, GNA and RCA (arrows). In contrast, in iii note the low and high reactivity of glial cells to GNA and RCA, respectively, resulting in a 'green-only' appearance (arrowheads). Scale bars 25 µm. (**C**) Surface levels of ConA, GNA, RCA and WGA reactive glycans in fibroblasts (COS, white; BHK, light grey; and CHO, dark grey) and in neurons (green). Mean ± SEM. N=27 to 132 cells in 2 experiments. *p<0.05; **p<0.01, ***p<1x10$^{-4}$; Kruskal-Wallis's multiple comparison test.

The following figure supplements are available for figure 1:

**Figure supplement 1.** Viable hippocampal neurons can be maintained in culture for more than two months.

*Figure 1 continued*

**Figure supplement 2.** Blocking the maturation of core-glycans with kifunensine increases the surface levels of ConA- and GNA-reactive glycans but decreases RCA- and WGA-reactive glycans.
**Figure supplement 3.** Intracellular ConA-, RCA- and WGA-reactive glycans are localized in distinct organelles in neurons.
**Figure supplement 4.** ConA-, RCA- and WGA-reactive glycans are localized in distinct organelles and display contrasting sensitivities to EndoH and PNGase in COS cells.

key synaptic proteins using their electrophoretic profiles after digestion by PNGase and EndoH, to identify core-glycosylated (EndoH-sensitive) proteins among the family of N-glycosylated (PNGase-sensitive) surface molecules (*Figure 3B*). We found that a substantial fraction of purified surface GABA_A receptor or AMPA-type and NMDA-type glutamate receptor subunits were core-glycosylated (EndoH-sensitive) (*Figure 3C*). More specifically, these proteins were either entirely core-glycosylated (e.g. GluN1 and GABA_AR β3), partially core-glycosylated (i.e. displayed both mature and core-glycosylated glycans on the same polypeptide chain, such as GluA1, and GluN2B) or were expressed as a mixture of proteins with either core-glycosylated or mature N-glycans (e.g. GluA2, GluA4). We note that while there are few reports of core-glycosylated sugars on functional proteins (*Hirayama et al., 2016*), these are typically associated with a few, select glycosylation sites thought to be inaccessible to Golgi enzymes. In contrast, here we show that all of the glycosylated sites of GluN1 (an obligatory subunit of NMDA receptors with 10 predicted N-glycosylation sites), GluA2 and GluA3 (4 predicted sites) are core-glycosylated. In contrast, TARP γ8, an AMPA receptor auxiliary protein that is thought to be co-trafficked with nascent receptors in the secretory pathway (*Tomita et al., 2003*) was completely insensitive to EndoH (*Figure 3C*) as is expected for a typical mature surface glycoprotein in non-neuronal cells (*Figure 3—figure supplement 1*). Similarly, the synaptic adhesion protein Neuroligin 1 (NLG1, *Figure 3C*) was also completely insensitive to EndoH. These results strongly indicate that synaptic proteins and receptors are processed and trafficked to the cell-surface through distinct mechanisms / secretory routes: a classical pathway that generates glycoproteins with mature N-glycans (e.g. TARP γ8 and NLG1), which in some instances are detected together with core-glycosylated sites on the same polypeptide (e.g. GluA1), as well as an unconventional route that generates proteins with only core-glycosylated profiles (e.g. GluN1 and GABA_AR β3).

## Hundreds of key surface neuronal membrane proteins are core-glycosylated

While both binding to ConA and digestion by EndoH rely on a similar biochemical substrate - a sufficient number of mannose residues in a given N-glycan – these two reagents are typically used experimentally to probe molecules that should reside in different cellular compartments: ConA is used to block the desensitization of AMPA and kainate receptors on the surface (*Reiner and Isacoff, 2014*; *Hoffman et al., 1998*; *Evans and Usherwood, 1985*) and EndoH is used to mark immature intracellular proteins in total cellular extracts (*Tucholski et al., 2013a*; *Rouach et al., 2005*). To assess whether ConA and EndoH might interact with the same proteins in neurons, we devised a two-round purification strategy to assess the EndoH-sensitivity of surface proteins interacting with ConA (*Figure 4A*). Consistent with their respective electrophoretic profiles after deglycosylation (*Figure 3C*), GluN1 and GluA2 were purified based on their interaction with ConA, whereas TARP γ8 was not detected (*Figure 4B*). As expected, the binding of surface GluN1 and GluA2 to ConA was abolished by competition with free mannose (*Figure 4C*). Finally we observed that prior digestion with EndoH also blocked GluA2 and GluN1 binding to ConA (*Figure 4D*), thus demonstrating that in hippocampal neurons, the binding of surface membrane proteins to ConA and their sensitivity to EndoH are equivalent. Thus EndoH-sensitive proteins – proteins that are classically regarded as intracellular and immature - are abundant at the neuronal surface.

How many surface proteins fall in this category and how important are they for neuronal function? To determine whether core-glycosylation is limited to a small set of neuronal surface proteins or

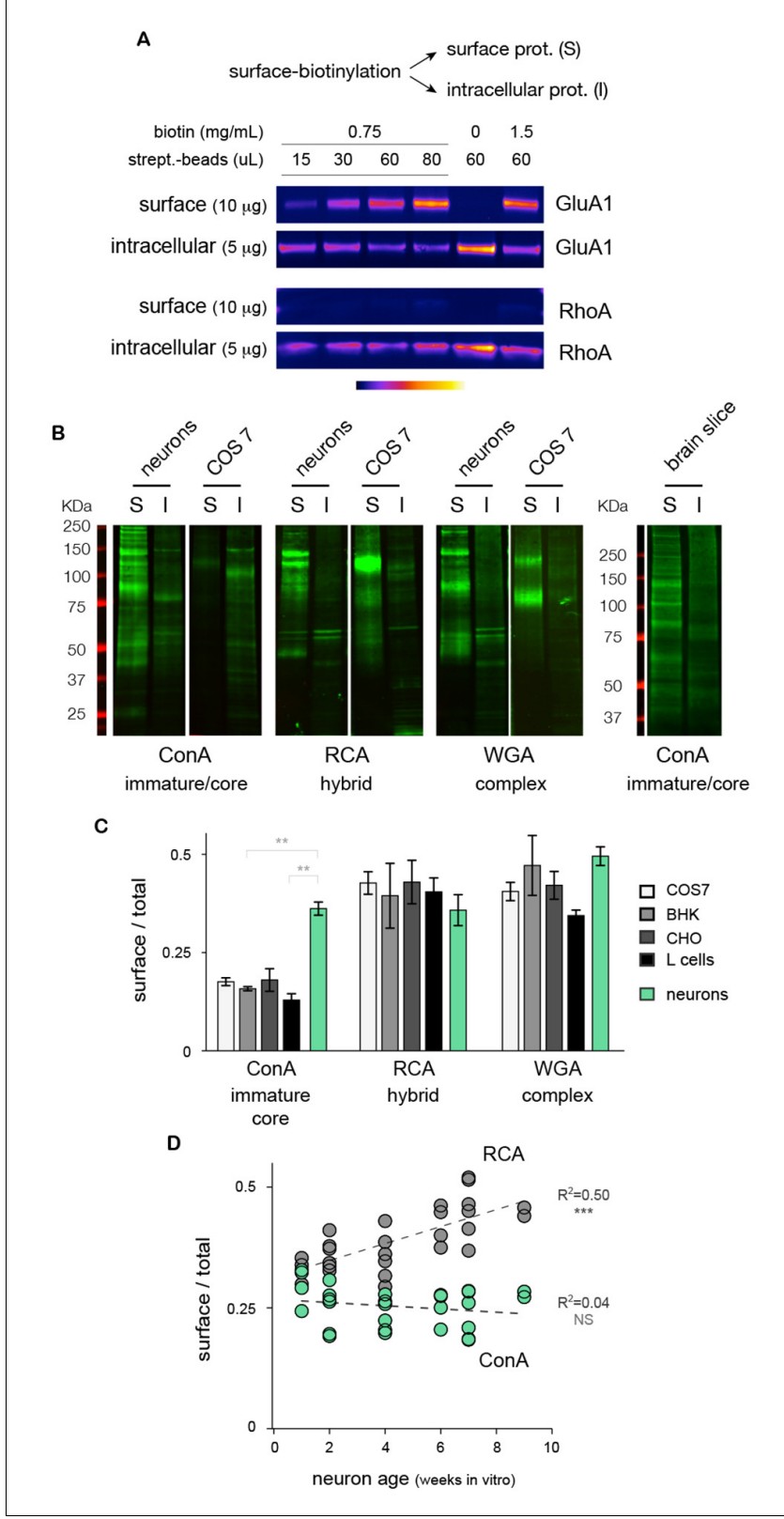

**Figure 2.** Core-glycosylated proteins are expressed at the neuronal surface at levels similar to hybrid and complex N-glycosylated proteins. (**A**) Isolation of surface ion channel (GluA1) versus cytoplasmic proteins (RhoA) after surface biotinylation and purification with varying amounts of NHS-SS-biotin and streptavidin-beads. (**B**) Surface (S) and intracellular (I) proteins prepared from neurons, COS7 cells and acute brain slices after labeling (far western

*Figure 2 continued on next page*

*Figure 2 continued*

blots) with ConA, RCA and WGA. For each condition, 15 and 5 µg of proteins were loaded for S and I fractions, respectively. Note the high levels of ConA reactive species in cultured neurons and in brain slices. (C) Relative surface expression (surface / total signal) of ConA, RCA and WGA reactive species in cultured neurons (green) and different non-neuronal cell types (COS, white; BHK, light grey; CHO, darker grey; L-cells, black) showing that the high surface expression of core N-glycans is specifically observed in neurons while other cell types show low levels of core glycosylation but higher levels of hybrid and complex glycosylation. Mean ± SEM and individual data points. N=4 in 2 experiments. * p<0.05; ** p<0.01, Dunn's multiple comparison test. (D) Relative surface expression of ConA- and RCA-reactive glycans in neurons from one to nine weeks in vitro, showing stable levels of surface core-glycosylation and a developmental increase in the surface expression of hybrid glycans. N= 27–28 dishes in 3 independent experiments. ***p<1x10$^{-4}$; Pearson's r test.

The following figure supplements are available for figure 2:

**Figure supplement 1.** The neuronal plasma membrane is intact after surface biotinylation.

**Figure supplement 2.** ConA-, RCA- and WGA-reactive glycans assessed by far-western blotting display distinct and specific sensitivities to PNGAse and EndoH.

---

represents a more widespread phenomenon, we used high-resolution mass spectrometry to identify neuronal surface proteins that interact with ConA (*Figure 4A,E and F*). To increase the specificity of our detection and obtain a high-confidence dataset, we used a label-free quantification (LFQ) approach to identify proteins that were consistently detected across experiments and whose binding to ConA was impaired by EndoH. In brief, plasma membrane proteins were first isolated after surface biotinylation and affinity purified with ConA, after ('background' control group) or without prior treatment with EndoH (target 'core-glycosylated' group). Purified proteins were fragmented with proteases and the resulting peptides were separated by HPLC, ionized and subjected to LFQ and fingerprinting by mass spectrometry. Surface core-glycosylated proteins were identified by focusing the analysis on peptides that showed the highest reproducibility and enrichment in the target group (see Materials and methods and *Figure 4—figure supplement 1*).

As expected, the resulting protein dataset was markedly enriched for secreted and plasma membrane transmembrane proteins with predicted N-glycosylation sites (*Figure 4—figure supplement 2*). To our surprise, we found that hundreds of transmembrane or secreted neuronal proteins were core-glycosylated (n=227 protein groups, 647 protein IDs; *Supplementary file 1A–1C*), including major ionotropic and metabotropic glutamate and GABA receptor subunits, synaptic adhesion proteins, neurotrophin receptors and voltage-gated ion channels (*Figure 4E*). Protein pathway and gene ontology analysis of our dataset compared to a control dataset (a proteome obtained from total neuronal lysates) (*Supplementary file 1D*) showed that multiple functional classes of glycoproteins related to axon guidance, synaptic transmission, synaptic plasticity and addiction were significantly overrepresented among the family of coreglycosylated surface proteins (*Figure 4F* and *Supplementary file 1E*). Of note, multiple terms associated with autoimmune diseases and cancer were also overrepresented in our dataset (*Figure 4F* and *Supplementary file 1E*). As with any affinity purification procedure, we cannot exclude the possibility that some false positive proteins are present owing to their strong interaction with surface proteins. We note, however, that we isolated surface AMPA receptor subunits from TARP γ8 based on their distinct glycosylation profiles (*Figure 4B*), even though these two proteins that are typically found in the same macromolecular complexes (*Schwenk et al., 2014*), thus suggesting a high level of specificity.

Taken together, the above data indicate that a large number of neuronal plasma membrane proteins display unconventional N-glycosylation profiles that are typically expressed at low levels in the plasma membrane of heterologous cells, suggesting that core-glycosylation plays a major physiological role in neurons.

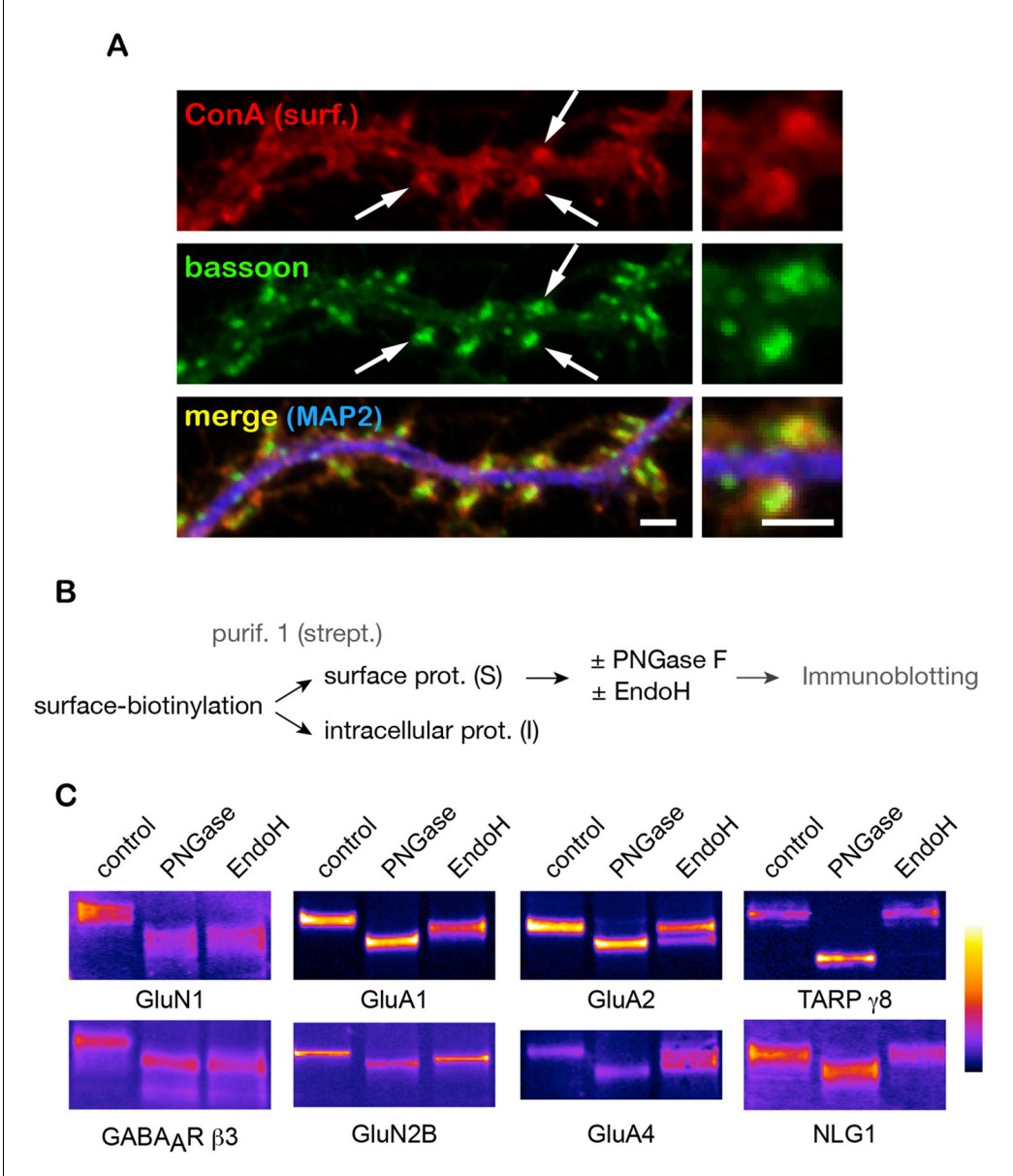

**Figure 3.** A number of key neuronal surface proteins display unconventional N-glycosylation profiles. (A–C) Synaptic localization of ConA reactive species and core-glycosylation of surface excitatory neurotransmitter receptors. (A) Fluorescent micrographs of cultured hippocampal neurons (DIV 26) after labeling of surface core-glycosylated proteins (ConA, red) and presynaptic terminals (bassoon immunoreactivity, green, shown together with MAP2-IR in blue) showing the presence of core-glycosylated proteins at synapses. Scale bar 5 µm. (B,C) Purification scheme (B) and migration profiles (C, immunoblots) of a group of surface glutamate receptors and associated proteins after PNGase and EndoH treatment. Note the complete EndoH sensitivity of GluN1 and GluA1, the partial sensitivity of GluA2 and GluA4 and the lack of sensitivity of TARP γ8 and neuroligin 1 (NLG1).

The following figure supplement is available for figure 3:

**Figure supplement 1.** Standard mature surface N-glycans are typically insensitive to EndoH.

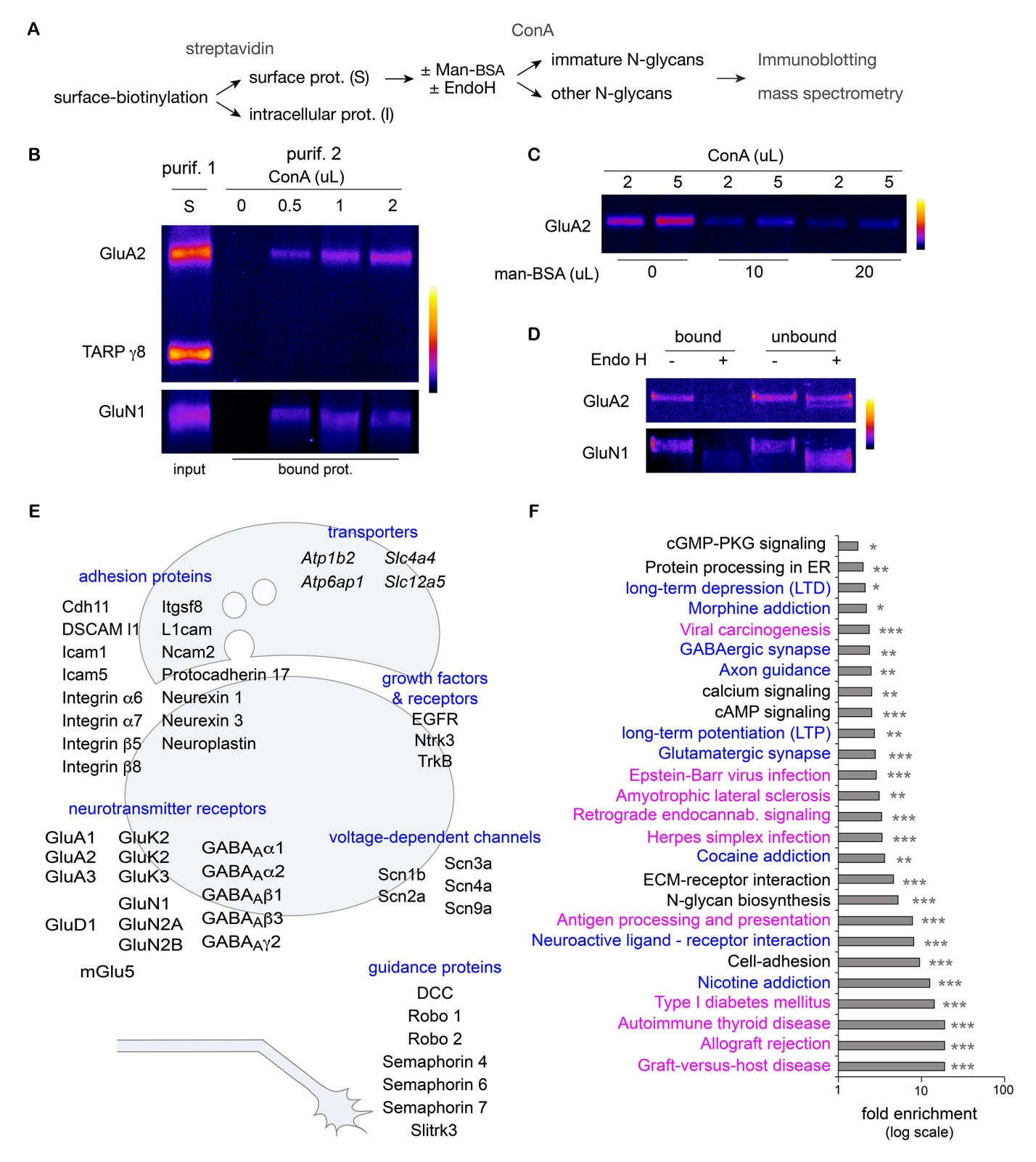

**Figure 4.** Hundreds of surface neuronal membrane proteins are core-glycosylated. Purification scheme (**A**) and immunoblots (**B–D**) showing the isolation of core-glycosylated N-glycans of the neuronal plasma membrane. (**B**) Presence of GluA2 and GluN1, but not TARP γ8, among proteins binding to ConA (vol. in µL). The lane 'S' shows GluA2, GluN1 and TARP γ8 immunoreactivity in the surface fraction that was used for the experiment. (**C**) Incubation of surface proteins with ConA in the presence of mannose-BSA (vol. in µL) abolishes GluA2 binding to ConA. (**D**) Bound and unbound

*Figure 4 continued on next page*

*Figure 4 continued*

fractions after purification with ConA in control conditions or after protein digestion with EndoH. Core-glycan removal with EndoH abolishes GluA2 and GluN1 binding to ConA. (E,F) Mass spectrometry of proteins purified as in A. (E) Schemes of a synapse and a growth cone highlighting examples of core-glycosylated surface neuronal proteins of particular physiological relevance for synaptic function and neuron development. (F) Fold enrichment of N-glycoproteins included in the indicated pathways in core-glycosylated surface proteins versus total proteins. Highlighted in blue and in purple are proteins of particular relevance for neuronal functions or autoimmune diseases and cancer, respectively. *p<0.05, **p<0.01, ***p<1x10$^{-4}$, hypergeometric test.

The following figure supplements are available for figure 4:

**Figure supplement 1.** Evaluation of mass spectrometry data part 1.

**Figure supplement 2.** Evaluation of mass spectrometry data part 2.

## Mature N-glycans are not required for dendritic development and maintenance

During neuronal development, the extension and elaboration of dendrites and axons places a high demand on neuronal secretory function to provide membrane, and thus an equally high demand on protein glycosylation. To address whether the presence of core-glycans at the neuronal membrane is specific to a particular developmental stage, we examined the relative surface expression of core and hybrid N-glycans throughout neuronal and synaptic development (over a nine week time-course). We found that while the surface expression of core-glycans remains constant, the surface expression of hybrid glycans progressively increases as neurons mature (*Figure 2D*), documenting a development-dependent regulation of neuron surface N-glycosylation. We next examined the dependence of dendritic development on N-glycans by treatment with tunicamycin (Tm), a drug that completely blocks all N-glycosylation by preventing the transfer of the N-glycan precursor to target proteins in the ER (*Figure 5A*). Dendritic growth and complexity were assessed by performing a Sholl analysis (*Figure 5C*) or by quantifying the total dendritic length and number of tips (*Figure 5D*). As shown in *Figure 5B*, we found that dendritic growth was severely impaired by a global blockade of N-glycosylation. To distinguish the contributions of mature vs. immature glycans to dendritic growth we used swainsonine (Sw), a selective blocker of Golgi type-II mannosidase Man2b1 and Man2b2 (*Tulsiani et al., 1982*), the enzymes that convert EndoH-sensitive proteins into EndoH-resistant species (a prerequisite for their subsequent maturation), and two other mannosidase inhibitors: kifunensine (Kf) and deoxymannojirimycin (DMJ) which inhibit ER or Golgi mannosidases acting upstream of Man2b1/Man2b2 (*Herscovics, 1999*) (*Figure 5A*). Surprisingly, we found that the immature N-glycans that remain following treatment with Kf, DMJ or Sw were sufficient to initiate and maintain dendritic growth (*Figure 5C–D* and *Figure 5—figure supplement 1*). To verify that these drugs had the expected effects on the surface expression of specific N-glycan subtypes, we assessed the surface expression of core, hybrid and complex glycans by far-western blotting (*Figure 5—figure supplement 2*). As expected, Kf, DMJ and Sw increased the surface levels of core-glycans while decreasing the levels of hybrid and, at least for Kf and DMJ, complex glycans. Thus, while N-glycosylation is necessary for dendritic development and maintenance, 'immature' N-glycans are sufficient to sustain these processes, which indicates that core-glycosylated proteins on the neuronal membrane surface are fully functional.

## Core-glycosylated proteins traffic to the neuronal surface through a non-canonical secretory pathway

As stated above, the trimming of the N-glycan mannose-core and the resulting loss of glycoprotein sensitivity to EndoH occurs in the Golgi apparatus (*Moremen et al., 2012*), raising the possibility that the neuronal core-glycosylated proteins that we detect on the plasma membrane may be trafficked to the cell-surface via a Golgi-independent mechanism. To address this, we used Brefeldin A (BFA) – a drug that is commonly used to disrupt the Golgi apparatus (*Klausner et al., 1992*) (*Figure 6A*) – and examined its effect on the surface expression of nascent proteins. To do so, neurons were pre-treated for with BFA (2.5 µg/mL; 1 hr), and metabolically labeled by bio-orthogonal non-canonical amino acid tagging (BONCAT) with azido-homo-alanine (AHA) for 120–150 min

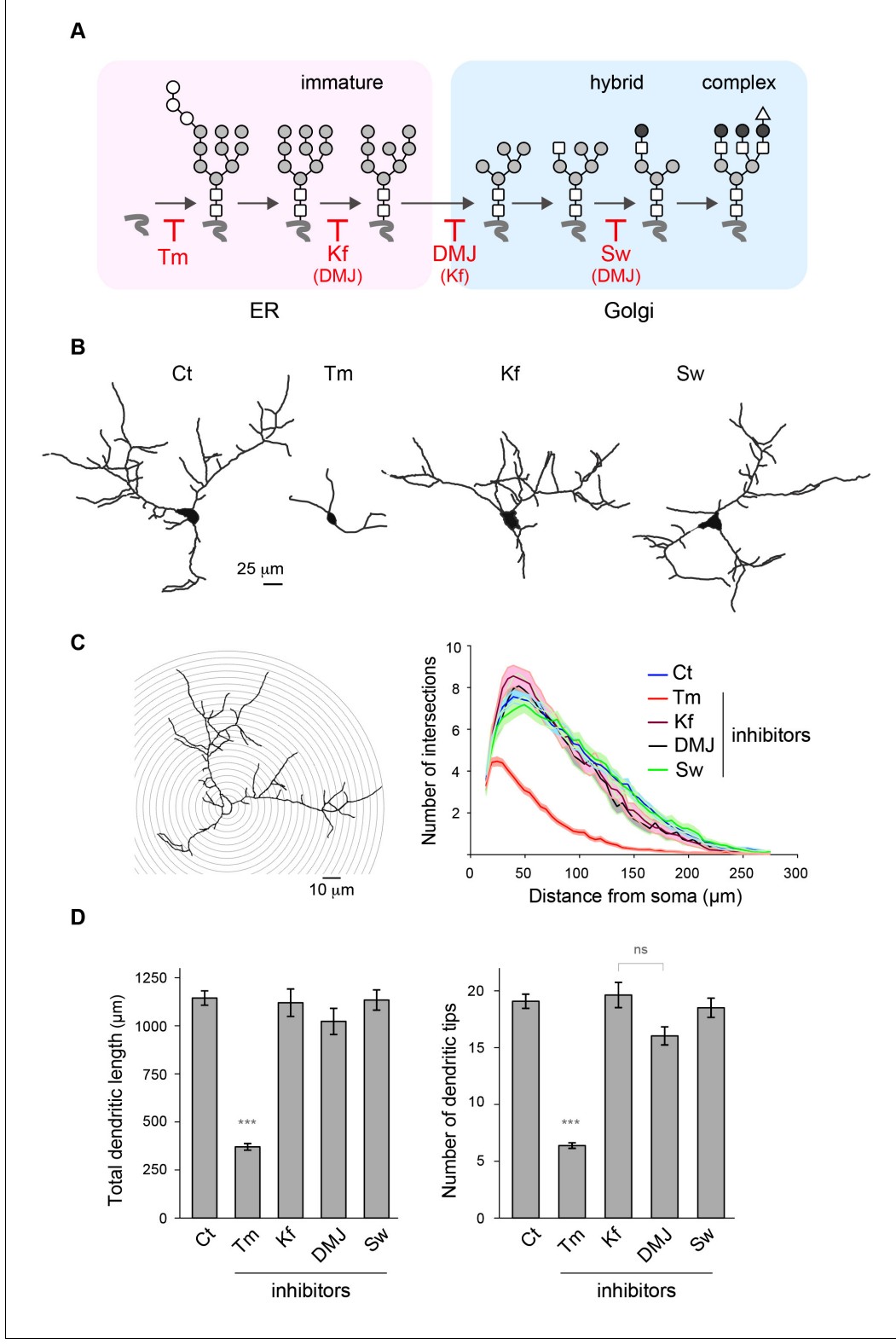

**Figure 5.** Mature N-glycans are not required for normal dendritic development. (**A**) Simplified representation of the N-glycosylation pathway and site of action of the glycosylation inhibitors tunicamycin (Tm), kifunensine (Kf), 1-deoxymannojirimycin (DMJ) and swainsonine (Sw). (**B–C**) Camera lucida drawings (**B**) and/or Sholl analysis (**C**) of the dendrites of hippocampal neurons (DIV 11) after a three day exposure to vehicle (DMSO, Ct), Tm (1.2 μM), Kf (5 μM), DMJ (75 μM) or Sw (88 μM). (**D**) Total dendritic length (left panel) and branch tip numbers (right panel) in
*Figure 5 continued on next page*

*Figure 5 continued*

neurons treated with vehicle (Ct), Tm, Kf, DMJ or SW. Mean ± SEM. N=31 to 137 cells in 2–6 independent experiments. ***p<1x10$^{-4}$; Kruskal-Wallis/Dunn's multiple comparison test. In **B–D**, note that while Tm strongly impairs dendritic growth and branching, blocking the maturation of N-glycans in the ER or the GA has no detectable effect on dendritic development.

The following figure supplements are available for figure 5:

**Figure supplement 1.** Other examples of dendrites after treatment with the drugs described in *Figure 5A*.

**Figure supplement 2.** The lack of effect of specific N-glycosylation inhibitors on dendritic development is not due to a lack of activity or to non-specific effects on the glycosylation pathways.

(*Dieterich et al., 2006*), or as a negative control methionine (*Figure 6—figure supplement 1*). Surface and intracellular proteins were then separated and quantified after surface-biotinylation and immunobloting (*Figure 6B and C*). As a positive control, similar experiments were performed in COS 7 cells. In neurons, BFA had no detectable effect on the levels of nascent surface proteins and slightly decreased intracellular nascent proteins (*Figure 6B*). In contrast in COS cells, BFA had no effect on intracellular proteins but markedly reduced the levels of nascent proteins at the plasma membrane (*Figure 6B*), as also observed for other cell types (*Davis and Mecham, 1996*). Thus, while BFA strongly reduced secretory trafficking in COS 7 cells, it had no detectable effect on the accumulation of nascent proteins at the neuronal surface under these experimental conditions (*Figure 6C*). We cannot rule out that BFA impairs secretory trafficking in neurons but that our method is not sensitive enough to detect this effect. Yet, our results indicate that secretory trafficking in neurons is markedly less dependent on the Golgi apparatus (GA) than in other cell types, providing an explanation for the atypical prominence of core-glycosylated proteins at the neuronal surface.

Do the complex and atypical (i.e. core-glycosylated) glycosylation profiles of candidate proteins reflect processing and lack of processing by the GA, respectively? To address this, we chose GluN1, GluA2 and TARP γ8 because of their physiological relevance and their respective complete, mixed (co-existence of EndoH sensitive and insensitive species) or absent core-glycosylation (*Figure 3C*). We found that exposure to BFA (5 µg/mL; 6–7 hr) significantly reduced the surface expression of TARP γ8 (*Figure 6D,E*)– as expected for a typical mature N-glycan (*Figure 3C*). The surface expression of GluA2 was also reduced by BFA, albeit to a lesser extent (*Figure 6E*). In contrast, the surface expression of GluN1 - a protein whose full surface complement is core-glycosylated – was insensitive to Golgi disruption (*Figure 6E*). Interestingly, the sensitivity to EndoH of these proteins was inversely correlated with their sensitivity to Golgi-disruption (*Figure 6E*). Importantly, as proteins with a faster turnover can be expected to respond more quickly to disruption of their biosynthetic pathway, protein stability must be taken into account in interpreting these results. We note, however, that previous studies have shown that GluA2 has a slower turnover ($t_{1/2}$ 1.95 days) than GluN1 ($t_{1/2}$ 1.61 days) (*Hanus and Schuman, 2013*), which indicates that the differential sensitivity of GluA2 and GluN1 to BFA cannot be accounted for by differences in stability. The stability of TARP γ8 is not known at present.

Thus, these data suggest that the surface expression of core-glycosylated glutamate ionotropic receptors is indeed due to a Golgi-independent secretory processing. Further supporting this view, our protein annotation analysis of the nascent proteins identified by mass spectrometry showed a significant enrichment for ER proteins, but not Golgi proteins, among the surface core-glycosylated proteins (*Figure 4—figure supplement 2*). Together with the relative paucity of canonical Golgi membranes as compared to other components of the secretory pathway in dendrites (*Hanus and Ehlers, 2008*; *Hanus et al., 2014*), the abundance of core-glycosylated proteins at the neuronal surface suggests that Golgi-by pass is surprisingly common for neuronal membrane proteins (*Torre and Steward, 1996*).

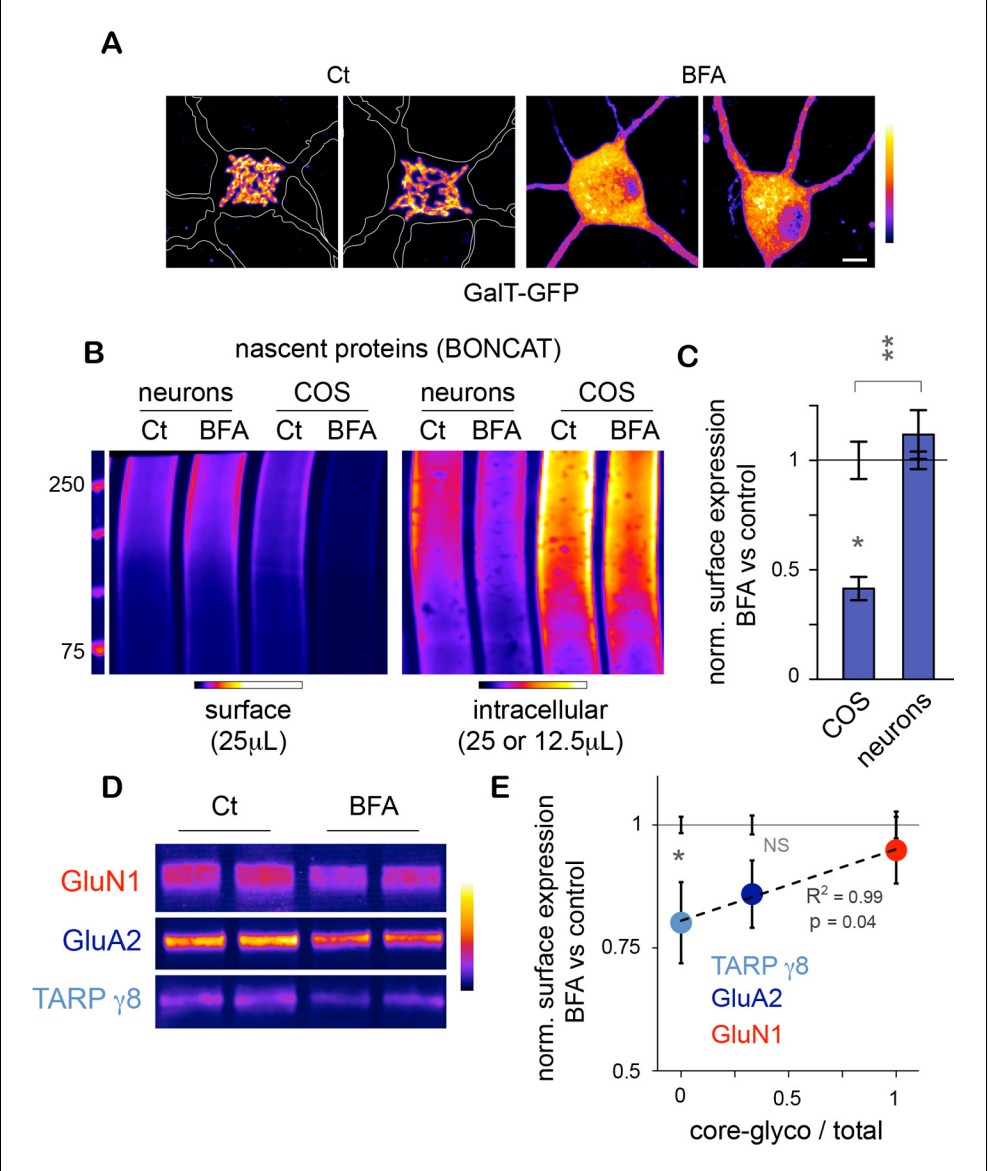

**Figure 6.** Unconventional secretory processing of core-glycosylated surface proteins. (**A**) GalT-GFP expression in neurons showing the morphology of the somatic Golgi apparatus after a 2 hr long exposure to vehicle (DMSO; control, Ct) or 2.5 µg/mL brefeldin A (BFA). Scale bar 5 µm. (**B–C**) Immunoblots (**B**) and surface or intracellular levels (**C**) of nascent proteins metabolically labeled for 120 to 150 min by bioorthogonal non-canonical aminoacid tagging (BONCAT) in COS cells and in neurons (DIV 25–26) in the absence (control or Ct) or in the presence of 2.5 µg/mL BFA. Note the strong effect of BFA on the surface expression of nascent proteins in COS cells and the relative lack of effect in neurons. N=5 to 7 dishes in 2–3 experiments. *p<0.05, **p<0.01; Dunn's multiple comparison test. See also *Figure 6—figure supplement 1*. (**D**) Immunoblot of surface GluN1, GluA2 and TARP γ8 after a 6–7 hr exposure to vehicle (Ct) or BFA, showing the reduced surface levels of TARP γ8, GluA2 and to a lesser extent GluN1 to the cell surface after disruption of the Golgi apparatus. (**E**) Sensitivity to BFA (BFA / control) as a function of core-glycosylation levels (core-glycosylated fraction / total). Note the selective effect of BFA on mature proteoglycans (low core glycosylation / total ratio). Mean ± SEM. N=10–11 in 4 experiments. *p<0.05, ANOVA/Tukey's multicomparison test.

The following figure supplement is available for figure 6:

**Figure supplement 1.** Nascent neuronal intracellular and surface proteins (BONCAT) after a 5-hr metabolic labeling with AHA or Methionine (Met) in the presence or absence of BFA.

## Core-glycosylation is sufficient to maintain synaptic transmission and modulates synaptic signaling

The coexistence of two forms of GluA2 on the cell surface – one form that presents only standard N-glycans and one form that presents only unconventional glycans – suggests that complex glycosylation profiles are not required for the expression of AMPA receptors on the cell-surface. To determine whether this was indeed the case, we used the glycosylation inhibitors described above to determine which glycosylation step(s) are required for the surface expression of GluA2. We found that whereas blocking N-glycosylation entirely with Tm markedly reduced GluA2 surface expression (*Figure 7—figure supplement 1*), blocking the maturation of core-glycans with Kf, DMJ or Sw had no detectable effect on GluA2 surface levels, despite clear effects (at least for Kf and Sw) on the glycosylation profile of the protein (*Figure 7—figure supplement 1*). Thus, the delivery of GluA2 to the neuronal surface requires N-glycosylation but does not require processing by Golgi glycosylation enzymes.

To determine more generally the dependence of synaptic receptors on mature glycans, we examined whether Kf impaired synaptic transmission using patch-clamp recording of spontaneous excitatory miniature postsynaptic AMPA/kainate currents. We found that blocking the maturation of core-glycans in the ER for 48 hr had no detectable effect on the frequency or the amplitude of synaptic currents (*Figure 7A,B*), thus strengthening the view that mature N-glycans are largely dispensable for synaptic transmission.

Do core-glycosylated proteins have specific functional properties? As a first step towards addressing this, we determined whether Kf, which inhibits processing beyond core-glycans, alters postsynaptic signaling by combining local glutamate uncaging and calcium imaging. We focused on AMPA-dependent signaling by monitoring postsynaptic calcium responses elicited by repetitive glutamate uncaging (~25 stimuli at ~2.5 Hz) in the presence of AP5 (an NMDA receptor blocker) (*Figure 7C–G*). This stimulation induced strong postsynaptic responses, which were completely blocked by the AMPA/kainate receptor blocker CNQX (*Figure 7D,E*). In blind recordings and analyses, we compared the responses of Kf-treated (for 48 hr prior to recording) and control neurons and discovered that, on average, the Kf-treated neurons exhibited responses with a shorter time to peak and a slight trend towards a larger decay (*Figure 7F*; *Figure 7—figure supplement 2*). It will be important for future studies to address how these parameters are mechanistically linked to increased core-glycosylation as multiple proteins including voltage-dependent calcium channels, calcium pumps and binding proteins (*Rose and Konnerth, 2001*) might be involve in shaping the intracellular calcium responses induced by AMPA receptor activation and postsynaptic depolarization. Nevertheless, our results show that core-glycosylation is sufficient to maintain normal synaptic transmission and may regulate the kinetics and magnitude of postsynaptic signaling.

## Accelerated turnover and regulated expression of core-glycosylated surface proteins

In native AMPA-type glutamate receptor complexes, the inclusion of the GluA2 subunit prevents the direct permeation by calcium (*Isaac et al., 2007*). Its presence or absence in synaptic receptor complexes is highly regulated, notably by protein synthesis (*Mameli et al., 2007*). We thus took advantage of the co-existence of standard (mature N-glycans) and core-glycosylated GluA2 subunits at the neuronal surface to directly compare the turnover of two distinct glycosylated forms of a well-studied and functionally important synaptic protein. Because the overall lifetime of a protein is a key determinant of how fast and by which mechanisms its levels can be tuned in a compartment specific manner (*Hanus and Schuman, 2013*; *O'Leary et al., 2013*), we compared the stability of core-glycosylated and mature glycosylated GluA2 with a chase assay after surface biotinylation (*Figure 8A–C*). The two forms of the protein were quantified by immunoblotting after separation by treatment with EndoH. Consistent with values measured for GluA1 in spinal cord neurons (*Mammen et al., 1997*), the overall stability of GluA2 (core + standard glycosylated forms) was on the order of tens of hours (15.8 ± 1.5 hr) (*Figure 8B*). However, the core-glycosylated pool of GluA2 exhibited a substantially shorter half-life than the standard form of the protein (3.4 versus 21.1 hr, respectively) (*Figure 8C*), demonstrating an accelerated turnover of the core-glycosylated form of the surface receptor.

GluA2 surface expression is also regulated during homeostatic synaptic scaling (*Gainey et al., 2009*). We thus determined whether modulation of synaptic activity also impacts the surface

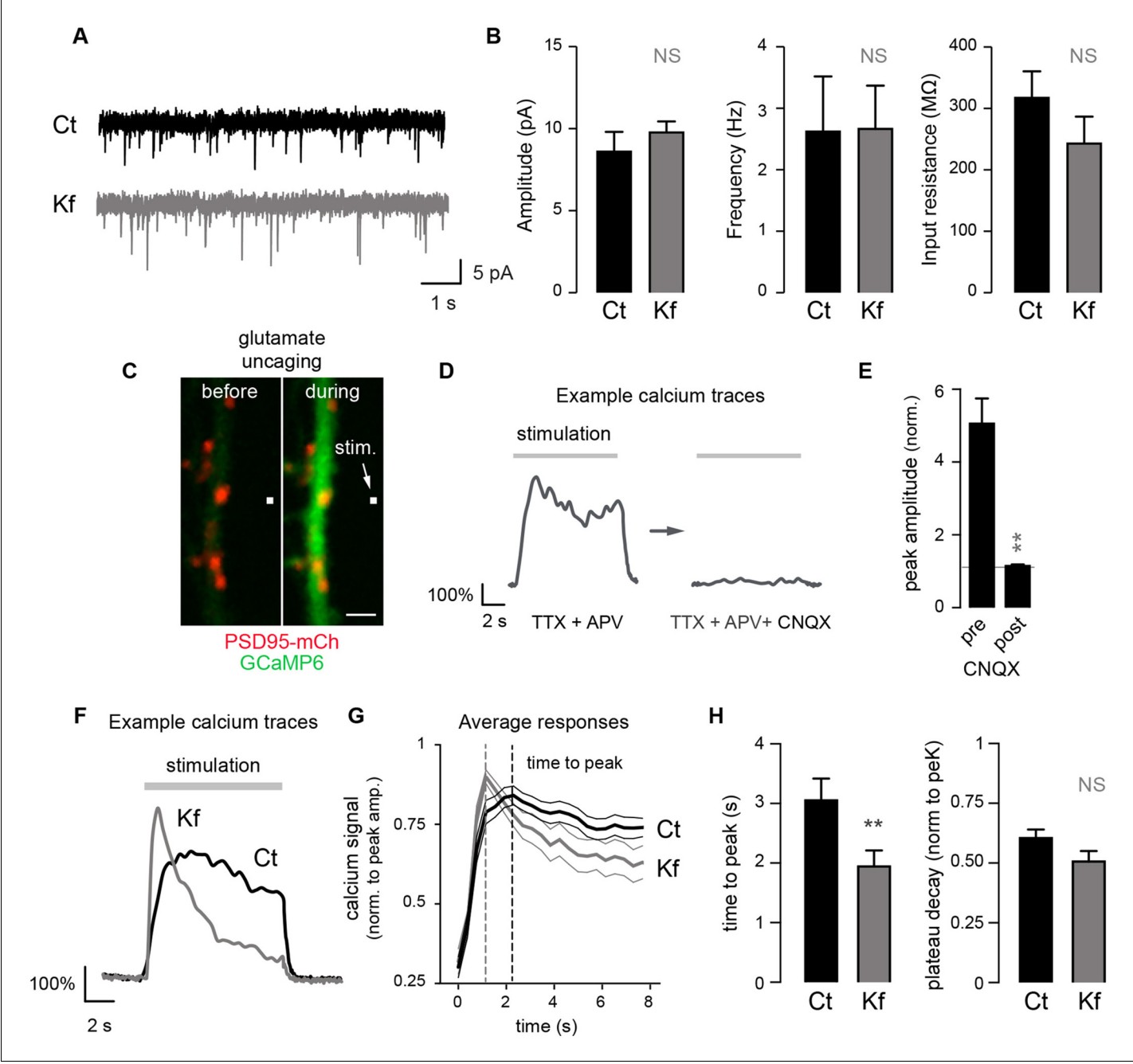

**Figure 7.** Blocking the maturation of N-glycosylation in the ER does not impair synaptic transmission and modulates the properties of postsynaptic AMPA receptors. (**A**) Example traces of miniature excitatory postsynaptic currents (mEPSCs) and (**B**) average mEPSC amplitude and frequency and input resistance (Mean ± SEM) in control (Ct) neurons (DIV 20–21) and in neurons exposed to Kf (48 hr prior to recording) showing the lack of effect of Kf on synaptic transmission. N = 10 and 7 cells for Ct and Kf, respectively, in 3 independent experiments. NS, not significant, Mann-Whitney's test. (**C**) Pictures of a synaptic marker (PSD-mCh, red) and a calcium indicator (GCaMP6, green, DIV14 neuron) before and after local glutamate uncaging at the position indicated by a white box (arrow). (**D**) Representative calcium traces and (**E**) average responses (peak amplitude normalized to baseline) induced by repetitive glutamate uncaging in individual dendrites (14 DIV neurons) before and after inhibition of AMPA receptors with CNQX. Note the complete block of the calcium signal by CNQX. N= 5 in 2 experiments. **p<10$^{-1}$; Mann-Whitney's test. (**F**) Example calcium trace, (**G**) average responses ± SEM (normalized to peak amplitude) and (**H**) time-to-peak and plateau decay in control neurons or in neurons exposed to Kf (48 hr prior to recording). In **G** and **H**, note the faster onset of postsynaptic responses in neurons treated with Kf. N= 29–31 (time to peak) or 22 (plateau decay) in 3 experiments. **p<10$^{-1}$; Mann-Whitney's test.

The following figure supplements are available for figure 7:

*Figure 7 continued on next page*

*Figure 7 continued*

**Figure supplement 1.** Core-glycosylation regulates GluA2 surface expression.

**Figure supplement 2.** Decay plateau (fraction of peak amplitude) as a function of time to peak (s) for synapses in control neurons (Ct, blue) or in neurons treated with Kf (pink).

expression of core-glycosylated N-glycans. Neuronal activity was either increased by blocking inhibitory synaptic transmission with the GABA$_A$ receptor blocker bicuculline (20 µM) or reduced with the ionotropic glutamate receptor blockers CNQX and AP5 (50 µM) for 20 hr and then the levels of

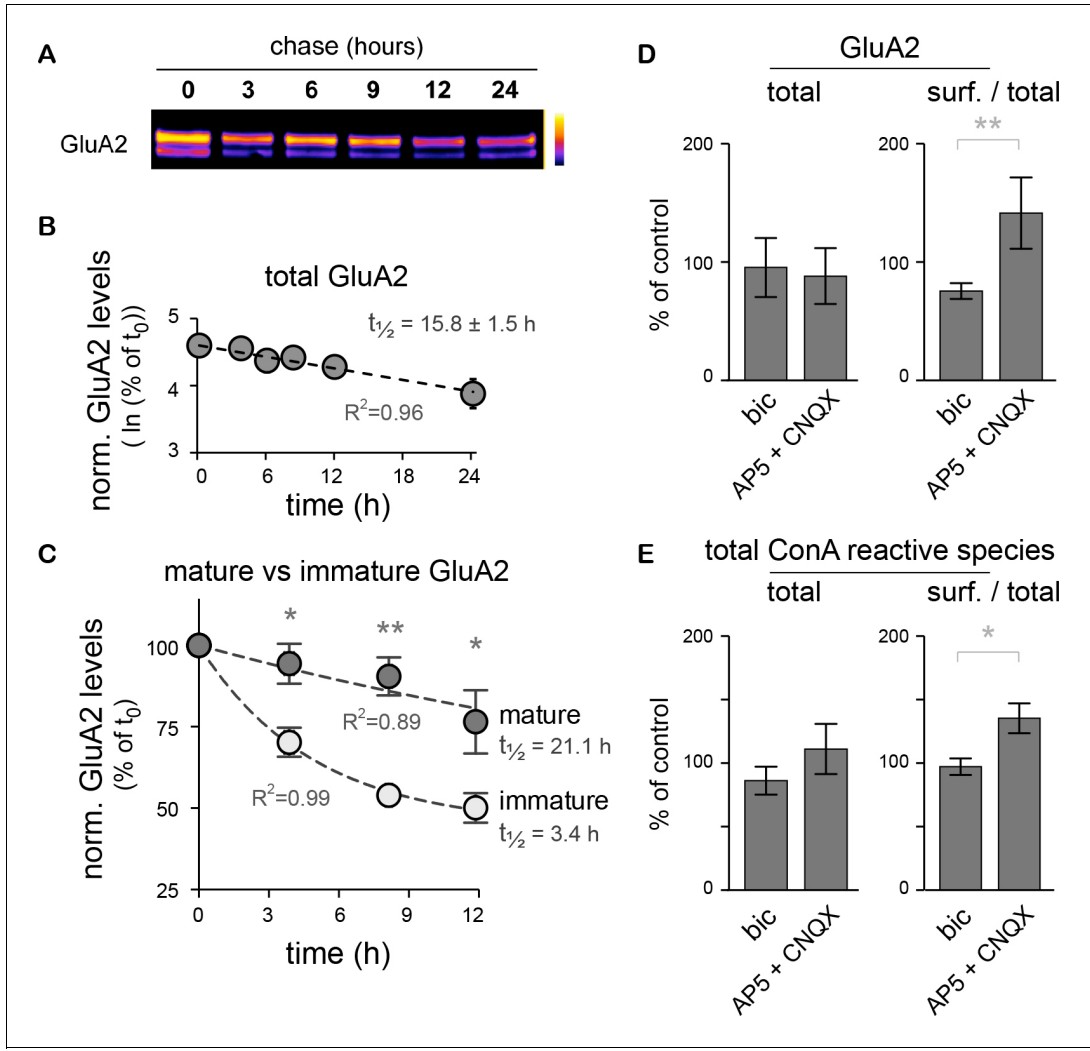

**Figure 8.** The core-glycosylation of surface proteins may accelerate their turnover and is regulated by synaptic activity. (**A**) Immunoblot of biotinylated-GluA2 at various chase-times (hours) after surface biotinylation. Surface proteins were deglycosylated with EndoH to allow the distinction of mature versus core-glycosylated receptors. (**B**) Decay of total biotinylated GluA2 levels (log scale, n=2–4 experiments). (**C**) Decay of mature versus core-glycosylated GluA2. Mean ± SEM; N=6–8 in 4 experiments. Note the shorter lifetime of core-glycosylated receptors. Protein half-lifes are indicated in hours. *p<0.05, **p<0.01, ANOVA/Tukey's multiple comparison test. (**D**–**E**) Total and relative surface expression (surf. / total) of GluA2 (**D**) or ConA-reactive glycans (**E**) after treatment with bicuculline (bic) or CNQX plus AP5 (CNQX + AP5). Bic and CNQX + AP5 levels normalized to control values. Mean ± SEM. N=7 in 4 experiments. *p<0.05, **p<0.01, Mann-Whitney test.

ConA-reactive species among surface and intracellular proteins were analyzed by far western blotting of total surface proteins. As expected (*Gainey et al., 2009*), we found that decreased synaptic activity increased GluA2 surface expression (*Figure 8D*). This was associated with an increase in the surface expression of total ConA-reactive (core-glycosylated) N-glycans (*Figure 8E*), thus indicating an activity-dependent regulation of core-glycosylated membrane protein trafficking.

## Discussion

It has been previously observed that, compared to other hexose-types, protein mannosylation was particularly pronounced in dendrites (*Torre and Steward, 1996*; *Villanueva and Steward, 2001*). Yet, it remained unclear whether mannose-rich N-glycosylated proteins are trafficked to the neuronal-surface. Here, we report that such 'immature' glycoproteins are abundant at the plasma membrane. We show that numerous synaptic adhesion proteins, surface neurotransmitter receptors, voltage-dependent ion channels and growth factor receptors present in the plasma membrane are core-glycosylated, thus displaying (mannose-rich) N-glycosylation patterns previously associated with nascent ion channels in the early secretory pathway (*Greger et al., 2002*; *Tomita et al., 2003*; *Sans et al., 2001*; *Rouach et al., 2005*).

While it is commonly believed that the maturation of N-glycans in the Golgi apparatus is required for the genesis of functional surface membrane proteins, we found that mature N-glycans are largely dispensable for proper dendritic development and spontaneous synaptic transmission. Interestingly, the core-glycosylation of surface proteins is regulated by synaptic activity and, as seen for GluA2, increases protein turnover. Altogether, these results point towards an important physiological role for core-glycosylation in neurons.

Although AMPA receptor subunits and TARPs are thought to be co-assembled and co-trafficked in the secretory pathway (*Zheng et al., 2015*), our results show that these proteins display distinct glycosylation profiles and distinct sensitivities to BFA. This indicates that these proteins are processed by distinct mechanisms and likely use distinct secretory routes (*Jeyifous et al., 2009*) to reach the plasma membrane. Intriguingly, the complete and selective loss of 'mature' N-glycosylated GluA2/3 in hippocampal neurons of TARP γ8 knockout mice results in a relatively modest decrease (34% ) in AMPA excitatory postsynaptic potential amplitude (*Rouach et al., 2005*). This suggests that core-glycosylated receptors, which account for 33% of total surface receptors, mediate 66% of synaptic transmission, which represents a relative contribution to synaptic currents (and an enrichment at synapses) that is ~4-fold higher. The present study thus challenges the notion that the core-glycosylated GluA2/3 subunits that remain unaltered in the TARP γ8 knockout are retained in the ER, and rather indicate that these receptors are localized at synapses and are functional both in wild type and in TARP γ8 knockout mice. Similarly, AMPA, kainate and GABA$_A$ receptors display abnormally increased or decreased EndoH sensitivities in schizophrenic patients (*Tucholski et al., 2013a*, *2013b*; *Mueller et al., 2014*). Our results question the interpretation that this merely reflects altered intracellular levels of immature receptors and rather suggest that these glycosylation defects impact functional synaptic receptors present on the plasma membrane.

Although dendritic Golgi outposts can be found in some neurons, most dendrites contain early secretory compartments (i.e. ER and ERGICs) but lack generic Golgi membranes (*Hanus and Ehlers, 2008*; *Hanus et al., 2014*). We previously showed that multiple postsynaptic signaling pathways control the local processing of nascent secretory cargo in dendrites (*Cui-Wang et al., 2012*; *Hanus et al., 2014*). However, it remained unclear whether and how nascent membrane proteins could be N-glycosylated in segments of dendrites lacking Golgi compartments. Thus far, few mammalian proteins have been shown to be trafficked to the plasma membrane via unconventional secretory processing (USP) (*Grieve and Rabouille, 2011*). The atypical prevalence of core-glycosylated proteins at the neuronal surface that we report here indicates that USP (e.g. Golgi by-pass or hypofunction of Golgi enzymes) is a much more widespread phenomenon than initially anticipated, and allows neurons to modulate the properties of key membrane proteins.

The correlation that we observed between the glycosylation status and the turnover of GluA2 is particularly interesting in this regard (*Hanus and Schuman, 2013*). The lifetime of a protein has a critical impact on its regulation in space and time. For example, a synaptic protein that is stable for multiple days has the time to potentially explore multiple synapses and go through several rounds of internalization and recycling before being degraded (*Ehlers, 2000*). It is thus hard to imagine how

the local dendritic synthesis of such a protein could lead to rapid and localized changes of synaptic composition and properties. On the other hand, a protein that is much less stable will likely be more efficiently regulated by a fine-tuning of its local synthesis and degradation. It is thus conceivable that the unconventional glycosylation of key proteins and the resulting decrease of their lifetime may tune the spatial length-scale over which local protein synthesis may functionalize synapses (*Govindarajan et al., 2011*).

In the recent years, our group and others have implemented and developed new tools and strategies to assess the distribution and translation of mRNAs (*Cajigas et al., 2012*; *Buxbaum et al., 2014*; *Wu et al., 2016*), the site of synthesis and the redistribution of specific nascent proteins (*tom Dieck, 2015*), and their dynamics in dendritic secretory organelles (*Cui-Wang et al., 2012*; *Hanus et al., 2014*). Here, we show that the glycosylation status of functional neuronal proteins is the result of distinct post-translational processing mechanisms that likely reflect the availability of secretory machinery in the different subcellular compartments that support protein synthesis. It will thus be interesting for future studies to investigate to what extent synthesis and secretory processing in the soma versus specific segments of dendrites determine protein glycosylation and hence their dynamics and function.

Previous studies in cancer cells have shown that surface proteins may be internalized, trafficked to the GA for further glycan maturation and sent back to the plasma membrane (*Snider and Rogers, 1986*). We cannot exclude that a similar mechanism exists in neurons and may convert core-glycosylated AMPA receptors into mature proteins. We note however that core-glycosylated GluA2 represents ~one-third of the GluA2 – an abundant neuronal protein – at the steady state. Owing to the paucity of dendritic Golgi membranes compared to endosomal compartments (*Ehlers, 2000*; *Cooney et al., 2002*), we find it unlikely that the large fraction of neuronal proteins identified here visit the Golgi following internalization. It will thus be interesting to investigate whether the rapid turnover of core-glycosylated synaptic receptors is due to a faster degradation or to another mechanism.

Given the prevalence of N-glycosylation in the brain (*Zielinska et al., 2010*), and its influence on membrane protein folding, trafficking, ligand-binding and ion channel conductivity (*Moremen et al., 2012*; *Scott and Panin, 2014*), it is clear that the atypical glycosylation of a large number of neuronal surface proteins is physiologically meaningful. In a broader context, it is worth noting that N-glycosylation is dysregulated in numerous human pathologies, and in particular in various cancers. Most notably, an increased branching of complex N-glycans is typically associated with a poor prognosis for breast and colon cancers in humans (*Fernandes et al., 1991*; *Seelentag et al., 1998*). Consistently, studies in mice show that Golgi-associated glycosyltransferases such as N-GlcNac, sialyl- and fucosyl-transferases are instrumental to tumor invasiveness (*Granovsky et al., 2000*; *Tsui et al., 2008*). The prevalence of core-glycosylated surface proteins in neurons may thus provide important insights on how N-glycans terminal branching is regulated and can be opposed. The immunological isolation of the brain likely plays a permissive role in surface proteins acquiring atypical glycosylation patterns in neurons, as those may otherwise trigger an immunological response. Indeed, multiple lines of evidence suggest that altered glycosylation profiles are important drivers of autoimmune diseases (*Rabinovich et al., 2012*; *Maverakis et al., 2015*). The enrichment of proteins involved in such pathologies among the core-glycoproteins that we identified thus indicates that investigating core-glycosylation in neurons may provide important cues on autoimmunity.

## Materials and methods

### Cell culture

Dissociated hippocampal neurons and cell lines (COS 7, BHK, CHO, L cells; obtained from the American Type Culture Collection) were prepared and maintained essentially as previously described (*Cui-Wang et al., 2012*; *Hanus et al., 2014*). Neurons were maintained for up to two months in vitro (*Figure 1—figure supplement 1*). Lack of cell line contamination with mycoplasma was checked by PCR (eMyco detection kit, Intron Biotechnology).

## Brain slices

Acute hippocampal slices were prepared from three week-old Sprague Dawley rats and the CA1 area carefully microdissected by hand as previously described (*Cajigas et al., 2012*).

## Plasmids and transfection

The VSVG-GFP, GalT-GFP, PSD95-mCh and pHluo-TM plasmids were described previously (*Cui-Wang et al., 2012*). GCaMP6-S (*Chen et al., 2013*) was purchased from Addgene (plasmid 40753). COS 7 and neurons were transfected with Extreme Gene 9 or Lipofectamine 2000 (Life Technologies), respectively, according to the manufacturer's instructions.

## Drug treatments

All drugs were used in the neuron maintenance medium. BFA (Sigma or Tocris), bicuculline, 6-cyano-7-nitroquinoxaline-2,3-dione (CNQX) and amino-5-phosphonovaleric acid (AP5) (Tocris) were used at final concentrations of 5 µg/mL, 20 µM, 50 µM, 50 µM, respectively. Tunicamycin (Tm), Kifunensine (Kf), deoxymannojirimycin (DMJ) and swainsonine (Sw) (Tocris) were used at final concentrations of 1,2 µM, 5 µM, 75 or 100 µM and 88 µM, respectively.

## Glycosydases

Peptide-N-Glycosidase F (PNGase, New England Biolab) and Endoglycosidase H (EndoHf, NEB) were used according to the manufacturer's instructions. In brief, proteins were denatured in PBS supplemented with 1% Triton, ~0.6% SDS and 50 mM DTT for 15 min at 75°C, and diluted (~1.5 fold) in sodium phosphate (50 mM pH 5.5 final) or sodium citrate buffer (50 mM pH 7.5) plus NP40 (or triton, ~1% final) for PNGase and EndoHf, respectively. Proteins were typically digested with 1000 (PNGase) or 3000 (EndoHf) units/ug protein at 37°C overnight.

## Lectins, antibodies and fluorescent streptavidine conjugates

The following lectins (fluorescein or biotin conjugates) were used for cytochemistry (CC) or far-western blotting (FWB) at the indicated final concentrations. Concanavalin A (biotin-ConA, Sigma; CC, 0.33 µg/mL; FWB, 1 µg/mL), galanthus nivalis agglutinin (biotin-GNA, Galab; CC, 1 µg/mL), RCA120-fluorescein (RCA120-FITC, Vector laboratories; CC, 0.7 µg/mL) and RCA (biotin-RCA, Vector laboratories; FWB, 1 µg/mL), wheat germ agglutinin (biotin-WGA, Sigma; CC 0.4 µg/mL; FWB 1 µg/mL), Streptavidine Alexa647 (Life Technologies, CC, 1 µg/mL), IRDye-streptavidin (Licor, FWB, 1:15,000). The following antibodies were used for immunocytochemistry (ICC) or immunoblotting (IB) at the indicated dilutions. Mouse anti-βactin (Sigma, IB, 1:10,000), mouse anti-bassoon (Enzo Life, ICC, 1:1000), rabbit anti-biotin (Cell signaling, ICC, 1:1000), rabbit anti-cacng8/stargazin (Millipore, IB, 1:1000), rabbit anti-GABA$_A$ receptor β3 subunit (Synaptic Systems, IB, 1:750), rabbit anti-GABA$_A$ receptor γ2 subunit (Synaptic Systems, IB, 1:1000), chicken anti-GFP (Aves Labs, IB, 1:5000), rabbit anti-GluA1 (Abcam, IB, 1: 1000), rabbit anti-GluA2 (Abcam, IB, 1:2000), rabbit anti-GluA4 (Synaptic Systems, IB, 1:1000), mouse anti-GluN1 (BD Pharmingen, IB, 1:1000), rabbit anti-GluN2A (Millipore, IB, 1:1000), rabbit anti-GluN2B (Millipore, IB, 1:1000), guinea pig anti-MAP2 (SYSY, ICC, 1:2000), mouse anti-MAP2 (Sigma, ICC, 1:1000), mouse anti-Neuroligin 1 (Neuromab, IB, 1:200), IRDye secondary antibodies (Li-Cor, IB, 1:15,000), goat anti-guinea pig-Alexa 647 (Life Technologies, ICC, 1:750), goat anti-mouse (GAM)-RRX (Jackson Laboratory, ICC, 1:1000), goat anti-rabbit-RRX and GAM-FITC (Jackson Laboratory, ICC, 1:800).

## Surface labeling and immunocytochemistry

For labeling of surface glycoproteins, cells were rinsed in ACSF (119 mM NaCl, 2.5 mM KCl, 1.3 mM MgSO$_4$, 2.5 mM CaCl$_2$, 1.0 mM NaH$_2$PO$_4$, 26.2 mM NaHCO$_3$, and 11.0 mM glucose) or in Hibernate A without phenol red (Brain Bits) and incubated with lectin biotin conjugates diluted in ACSF or Hibernate A, for 10 min at room temperature. After washes, cells were fixed in 4% PFA, blocked in PBS supplemented with 1% fish skin gelatin (Sigma) for 15 min and incubated for 30 min with streptavidin-Alexa647 in the same buffer. For co-immunolabeling, cells were incubated live with ConA-biotin, fixed and permeabilized in 0.2% Triton in PBS for 15 min, blocked in a buffer containing 10% goat serum (Life Technologies) and 3% fish skin gelatin for 30 min and incubated with primary (anti-

bassoon, anti-biotin and anti-MAP2) and secondary antibodies diluted in a 1:2 dilution of the same blocking buffer.

## Image acquisition and analysis

Confocal imaging was performed with a 40x 1.4 NA objective on Zeiss LSM780 or LSM880 laser point scanning confocal microscopes. Cell average fluorescence was quantified in Metamorph (Molecular Devices) using Z-stack maximal intensity projections.

## Surface-biotinylation and pulse chase experiments

Surface biotinylation was performed essentially as described previously (*Cui-Wang et al., 2012*; *Hanus et al., 2014*; *Ehlers, 2000*). In brief, cells were rinsed in ACSF or E4 buffer (120 mM NaCl, 3 mM KCl, 15 mM glucose, 10 mM HEPES, 2 mM CaCl$_2$, 2 mM MgCl$_2$ CaCl$_2$) and incubated with 0.8 to 1 mg/mL NHS-SS-biotin (Thermo) in the same buffer for 7 min at room temperature. Cells were then rinsed and quenched in ACSF or E4 supplemented with 10–20 mM L-lysine, scraped in PBS supplemented with L-lysine and, when appropriate, protease inhibitors (Calbiochem). Cell pellets harvested after centrifugation were then either stored at −80°C or directly lyzed. Pulse chase experiments were performed essentially as described previously (*Cui-Wang et al., 2012*; *Mammen et al., 1997*). In brief, cells were biotinylated, washed in ACSF supplemented with 0.1% BSA, and put back in fresh (cell lines) or conditioned (neurons) maintenance medium and kept at 37°C for varying times. Slices were surface-biotinylated in oxygenated ACSF at 4°C for 20 min, quenched in L-lysine at 4°C for 10 min, homogenized and then lyzed as descried above. Surface (biotinylated) proteins were eluted from streptavidin agarose or magnetic beads (Thermo) by reduction of S-S-biotin with 50 mM DTT (in PBS supplemented with ~1% Triton and ~0.6% SDS) for 15 min at 75°C, resulting in the complete removal of biotin from surface proteins (*Figure 4B* and *Figure 6—figure supplement 1*).

## Immunoblotting and far-western blotting

Immunoblotting was performed essentially as described previously using far-red fluorescent dyes and a Licor Odyssey scanner (*Cui-Wang et al., 2012*). For far-western blotting, lectin biotin-conjugates, antibodies and streptavidin were diluted in PBS-Tween without the addition of blocking agents. Protein levels in bands of interest were quantified in Image Studio (Licor) or ImageJ (NIH).

## Quantification of protein surface expression after surface biotinylation

Known amounts of proteins were separated into a surface (biotinylated) and an intracellular fraction (remaining supernatant) and processed in parallel for immunoblotting or far western blotting. The total fluorescent intensity of individual bands (immunoblotting) or smear of proteins (far western blotting with lectins, or BONCAT, e.g. between ~250 and ~ 70–25KDa) was then quantified and protein relative surface expression calculated as follows: with S, I the total fluorescence intensity and 1/a, 1/b sample dilution factors in surface and intracellular fractions, protein relative surface expression was calculated as the ratio: $S / (S + bI/a)$.

## Purification of surface N-glycans

Surface (biotinylated) proteins were eluted from streptavidin-agarose beads at 70°C with 50 mM DTT for 15 min and incubated in the absence (group A) or in the presence (group B – background) of EndoHf (NEB) overnight at 37°C. EndoHf was then heat-inactivated at 80°C for 25 min. Proteins were then incubated with ConA biotin-conjugate in PBS supplemented with 1% Triton x100 and 0.1% SDS for 3 hr at 4°C in the absence, or, for control experiments, in the presence of BSA-mannose (Vector laboratories) and purified with high-capacity streptavidin-agarose or streptavidin-magnetic beads (Thermo).

## Mass spectrometry

For each sample group A (target group: surface proteins binding to ConA without prior treatment with EndoH) and B (background group: surface proteins binding to ConA after treatment with EndoH), 2 independent biological replicates (surface proteome preparations from separate primary neuron preparations), 2 experimental replicates (replicate affinity purifications and digestions for

each surface preparation) and 3 to 4 technical replicates (replicate LC-MS runs on identical peptide preparations) were analysed, so a total of 13–14 replicates per sample.

Proteins were incubated in 6 M urea, 2 mM DTT, alkylated using 5 mM iodoacetamide and sequentially digested overnight using LysC (1:15 protease:protein ratio). After bringing the urea concentration to a final concentration of 3 M, the samples were incubated with Trypsin (1:15) overnight. The crude peptide mixtures were purified using $c_{18}$-ZipTips (Millipore), dried using a *SpeedVac* and stored at −80°C. Four sets of proteolytic fragments (two technical replicates for 2 independent neuronal preps) were loaded 3–4 time each on reverse phase HPLC columns (trapping column: particle size <2 μm, $C_{18}$, L=20 mm; analytical column: particle size <2 μm, $C_{18}$, L = 50 cm; ThermoFisher Scientific) using a nano-UPLC device (Dionex Ultimate 3000 RSLC, ThermoFisher Scientifc), and eluted in binary solvent gradients (buffer I: 5% acetonitrile, 95% water, 0.1% formic acid; buffer II: 80% acetonitrile, 20% water, 0.1% formic acid). Typically, gradients were ramped from 5% to 55% buffer II within 200 min at flow rates of 300 nl/min. Peptides eluting from the column were ionised 'online' using a FlexIonSource (Thermo) and analyzed in a hybrid ion trap mass spectrometer (Orbitrap Elite, ThermoFisher Scientific). Sequence information was obtained by computer-controlled, data-dependent switching to $MS^2$ mode (TOP15, FT-IT-mode) using collision energies based on mass and charge state of the candidate ions. Proteins were identified by matching the derived mass lists against a NCBI or 'refseq' database (downloaded from http://www.ncbi.nlm.nih.gov) on a local Mascot server (Matrix Sciences, United Kingdom). In general, a mass tolerance of 2 ppm for parent ions and 0.5 Da for fragment spectra, two missed cleavages and oxidation of methionine as a variable modification and carbamidomethylation of cysteine as a fixed modification were selected as matching parameters in the search program. For quantitative evaluation, the dataset was processed using the MaxQuant software package, Perseus and custom scripts in Matlab (https://github.molgen.mpg.de/MPIBR-Bioinformatics/AtypicalNeuroNGlycans).

We then used a high-stringency label-free quantification (LFQ) approach to identify proteins that were significantly enriched in samples A (target, core-glycosylated) over samples B (i.e. whose binding to ConA was impaired by EndoH and thus represent "background). The efficiency of EndoH was high but not absolute, resulting in B samples consisting of missing values and low-intensity peptides. Although protein abundance was clearly higher in A than in B, this bias (zero values for B sample peptides) complicated the use of average peptide intensities as in a typical LFQ approach. We thus developed a peptide based-strategy and sorted peptides according to their repeatability across experiments (intraclass correlation (ICC) index between biological replicates x ICC index between technical replicates) and their enrichment in A or B: ([average intensity in A – average intensity in B] / max intensity in A or B). Peptides were then separated by hierarchical clustering based on Euclidian distance (0.2 threshold in the distance matrix). Peptides with the highest enrichment and repeatability were clearly separated as a 'natural' cluster (*Figure 4—figure supplement 1A*), and were used to define our high-confidence 'core-glycosylated' proteins: resulting in the identification of 227 protein groups and 647 protein IDs (*Supplementary files 1A and 1C*).

As expected, the retrospective analysis of all the peptides that corresponded to these proteins showed a higher enrichment in A and a higher repeatability than the rest of the 'background' proteins (*Figure 4—figure supplement 1B and C*). As an additional validation, we crossed referenced these proteins with those identified using a standard LFQ approach. Here, we selected proteins based on peptides that were detected in at least 10 out of 13–14 'A' replicates, and whose average peptide intensity was at least 1.3 times higher (Kruskal Wallis test) in A than in B samples. 88.4% of the protein groups that we identified with our peptide-based strategy were also detected by this approach (*Supplementary file 1B*).

To compare our dataset to the full hippocampal proteome, we selected proteins identified by both mass spectrometry (*Sharma et al., 2015*) and mRNA deep sequencing in mouse hippocampi (*You et al., 2015*), yielding 19,690 proteins. Likely owing to the double purification procedure that was used to purify surface core-glycosylated proteins, a few proteins present in our A and/or B sets were not found among these proteins and were thus added to the later list to generate our final input dataset (*Supplementary file 1D*).

Protein topology, N-glycosylation sites and subcellular localization were defined with Signal IP and TMHMM (*Sprenger et al., 2008*), NetNGlyc1.0 (http://www.cbs.dtu.dk/services/NetNGlyc/) and CELLO (*Yu et al., 2006*), respectively. For gene pathway and ontology analyses (*Figure 4* and *Supplementary file 1E*), overrepresented protein functional families were determined using the

Kyoto Encyclopedia of Genes and Genomes' (KEGG) pathway database (*Kanehisa et al., 2012*) or Gene Ontology annotation (*Ashburner et al., 2000*) using full protein lists, or subclasses determined according to expected topology (soluble intracellular versus secreted + transmembrane proteins) or predicted N-glycosylation sites.

## Distinction of core and hybrid N-glycans

ConA binds with high affinity to terminal α-linked mannoses, thus preferentially to core glycans but may, in theory, also bind to hybrid N-glycans. In this study, we can clearly distinguish between these two types of N-glycans: as exemplified in *Figures 1A* and *5A*, hybrid N-glycans are typically recognized by RCA. However, as shown in *Figure 2—figure supplement 2*, RCA-binding proteins are EndoH *insensitive* in hippocampal neurons, in contrast to ConA binding proteins that are EndoH sensitive. Further, as shown in *Figure 1—figure supplement 2* and *Figure 5—figure supplement 1*, RCA-reactive proteins respond completely differently from ConA-reactive species to Kf, DMJ and Sw. We cannot exclude that rare hybrid and yet unrecognized glycans that are Endo-H sensitive and selectively bind ConA but not RCA may be found in neurons. Yet, our data strongly indicate that core-glycans are atypically abundant at the neuronal surface.

## Quantification of dendritic morphology

Hippocampal neurons were transfected with GFP at DIV7 and were treated with vehicle (control, DMSO), Tm, Kf, DMJ, Sw at DIV8, fixed at DIV11 and immunolabeled for the somatodendritic marker protein MAP2. Dendrite morphology was assessed basically as previously described (*Cui-Wang et al., 2012*) using the Simple Neurite Tracer plugin (*Longair et al., 2011*; *Ferreira et al., 2014*) in ImageJ/Fiji (NIH). The experimenter was blind to the experimental conditions tested both during image acquisition and analysis.

## Protein metabolic labeling by bio-orthogonal non-canonical amino acid tagging (BONCAT)

Cells were preincubated for 1 hr with 2.5 ug/mL BFA (or vehicle, DMSO) and then metabolically labeled with 4 mM azido-homo-alanine (AHA) (*tom Dieck, 2015*) for 120 to 150 min (or for control for 5 hr with 4 mM AHA versus methionine, *Figure 6—figure supplement 1*) in Neurobasal without methionine supplemented with glutamax and B27. Cells were then surface biotinylated and surface (S) and intracellular proteins (I) separated as described above. S and I proteins (50 μg protein/lysate equivalent) were then precipitated in cold acetone. Precipitated samples were resuspended in 120 μL of PBS (pH 7.8) supplemented with 0.07% SDS and 0.1% triton and protease inhibitors, cleaned on desalting columns (PD Spin trap G25) and were then biotinylated by azide-alkyne cyclo-addition ('CLICK' reaction) as previously described (*tom Dieck, 2015*), immunoblotted and detected with an anti-biotin antibody.

## Calcium imaging and focal glutamate uncaging

DIV10 neurons were transfected with PSD-mCh and GCaMP6 as described above and treated with Kf for 48 hr. Neurons were then washed and monitored at 37°C in standard E4 medium (150 mM NaCl, 3 mM KCl, 15 mM glucose, 10 mM HEPES, pH 7.4) supplemented with 3 mM $CaCl_2$, 0.5 mM $MgCl_2$ and 50 μM AP5 and 2.5 mM MNI-glutamate (Tocris). Confocal imaging was performed using a 60x 1.4 NA objective on a Zeiss Observer Z1 inverted microscope equipped with a CSUX1 spinning disk unit (Yokugawa, Inc), an EM-CCD camera (Evolve 512, Photometrics), and a custom diode-laser illumination module (3I Intelligent Imaging Innovations, Inc). MNI-glutamate was uncaged by local irradiation at diffraction limited spots with a tunable 2-photon laser set at 720 nm (Chameleon Ultra, Coherent) coupled to a high speed x,y scanner (Vector). Dendrites were stimulated at individual synapses identified by PSD-mCh fluorescence. Imaging was done in alternance with 26 uncaging pulses delivered at 2.6 Hz. The peak amplitude and time to peak of stimulated calcium responses (GCaMP6 fluorescence) were determined after normalization to baseline by detection of fluorescence maxima. Responses decay and plateau were calculated after normalizing responses to peak values and fitting of the resulting decay plots with a mono-exponential function using Prism (Graph-Pad). The experimentalist was blind to experimental conditions during data acquisition and analysis.

## Electrophysiology

Whole-cell recordings were performed in DIV 20–21 neurons after a ~48 hr exposure to Kf. Neurons were held at −70 mV in voltage clamp and mEPSCs were recorded for at least 10 min using an Axopatch 200B amplifier. The extracellular solution contained (in mM) 140 NaCl, 3 KCl, 10 HEPES, 2 $CaCl_2$, 1 $MgSO_4$, 15 glucose, 0.002 TTX, 0.04 bicuculline and 0.05 AP5 (pH 7.4). Recording pipettes, with resistances between 3–8 MΩ, were filled with a solution containing (in mM) 120 Potassium gluconate, 20 KCl, 10 HEPES, 2 $MgCl_2$, 0.1 EGTA, 2 $Na_2$-ATP and 0.4 $Na_2$-GTP (300 mOsm/l, pH 7.2). Data were analyzed offline with a template-matching algorithm (*Guzman et al., 2014*) using Stimfit (*Guzman et al., 2014*) or in Matlab using a custom script. Selected mEPSC events were individually screened with an amplitude threshold of >5 pA and an exponential decay. The experimenter was blind to the experimental conditions tested during both acquisition and analysis. One outlier cell in the control group had a mEPSC frequency close to 4 standard deviations higher than average and was excluded from the data set.

## Statistics

Data are presented as means ± SEM unless otherwise indicated. The number of measured values and independent experiments used for quantification are indicated in the text or in the figure legends. Mann Whitney non-parametric test was used to compare two means. Data normality was assessed with Shapiro Wilk's or Kolmogorov Smirnov's tests. When data passed normality test, one-way ANOVAs and post hoc Tukey or Sidak's multicomparison tests were performed to determine whether significant differences existed among all or preselected pairs of means (i.e. control versus BFA) across multiple conditions (i.e. N>2). Otherwise, multiple comparisons were assessed with Kruskal-Wallis and Dunn's multiple comparison tests. Hypergeometric test was used to assess the significance of relative frequencies in total or subpopulation of proteins (e.g. core-glycosylated versus total proteins). See Statistical reporting in *Supplementary file 1F*.

## Acknowledgements

We thank Ina Bartnik, Nicole Fürst, Dirk Vogel, Anja Staab, Christina Thum and Imke Wüllenweber for excellent technical assistance. We thank Stuart EH Moore for insightful discussions. Work in the laboratory of EMS is supported by the Max Planck Society, the European Research Council, DFG CRC 902, 1080, and the DFG Cluster of Excellence for Macromolecular Complexes. CH is supported by a Marie Curie career integration grant.

## Additional information

### Funding

| Funder | Grant reference number | Author |
| --- | --- | --- |
| European Commission | Marie Curie Career Integration Grant 303818 | Cyril Hanus |
| European Research Council | Advanced Investigator Award | Erin M Schuman |
| Deutsche Forschungsgemeinschaft | Cluster of Excellence for Macromolecular Complexes | Erin M Schuman |
| Deutsche Forschungsgemeinschaft | CRC 902 | Erin M Schuman |
| Deutsche Forschungsgemeinschaft | CRC 1080 | Erin M Schuman |

The funders had no role in study design, data collection and interpretation, or the decision to submit the work for publication.

### Author contributions

CH, SG, LK, A-SH, JDL, Conception and design, Acquisition of data, Analysis and interpretation of data, Drafting or revising the article; HG, BA-C, Conception and design, Acquisition of data, Analysis

and interpretation of data; GT, EMS, Conception and design, Analysis and interpretation of data, Drafting or revising the article; SS, Acquisition of data, Analysis and interpretation of data

### Author ORCIDs
Erin M Schuman, ⓘ http://orcid.org/0000-0002-7053-1005

### Ethics
Animal experimentation: We hereby certify that all the experiments involving animals (i.e. postmortem tissue removal as defined in the § 4(3) of German animal welfare act) that were done in relation to our manuscript entitled "Unconventional secretory trafficking diversifies the properties of neuronal ion channels" were carried out in accordance with the European directive 2010/63/EU, the German animal welfare act, and the guidelines of the Federation of Laboratory Animal Science Associations (FELASA) and the Max Planck Society.

## Additional files

### Supplementary files
• Supplementary file 1. Tables. (A) Core-glycosylated surface proteins (protein groups and protein IDs). (B) Core-glycosylated surface proteins (protein groups cross-referenced with typical LFQ). (C) Core-glycosylated surface proteins (protein IDs, predicted topology, subcellular localization and N-glycosylation sites). (D) Input proteins. Proteins in total hippocampal extracts and/or in A or B groups displayed as in Table 3. (E) KEGG pathways. Top 25 functional classes for all proteins or predicted glycoproteins in core-glycosylated surface proteins versus total proteins. (F) Statistical reporting. Results of the statistical tests used for the indicated figures.

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
