## [Decision Letter]

Thank you for resubmitting your work entitled "Unconventional secretory processing diversifies neuronal ion channel properties" for further consideration at *eLife*. Your revised article has been favorably evaluated by Eve Marder (Senior editor), a Reviewing editor, and three reviewers.

The manuscript has been improved but there are a few remaining issues that need to be addressed before acceptance, as outlined below:

The reviews wonder why the blots of BONCAT samples in Figure 6 and Figure 6—figure supplement 1 look so different. Particularly the left panel in Figure 6 appears unusual – the strongest signal is at the edges of the lane. There is also a question about the combination of surface biotinylation and BONCAT. For the experiment shown in Figure 6, surface proteins were biotinlylated for the separation of surface and intracellular proteins, it is assumed by affinity purification, then again biotinylated in the BONCAT procedure, and then separated by SDS-PAGE and immunodetected by Western blotting with anti-biotin antibodies. This means, that the proteins analyzed in the surface protein fraction are biotinylated irrespective of the BONCAT treatment. Does this not confound the analysis and the interpretation that nascent proteins are analyzed specifically? Please address this issue in the resubmission.

[Editors’ note: a previous version of this study was rejected after peer review, but the authors submitted for reconsideration. The first decision letter after peer review is shown below.]

Thank you for submitting your work entitled "Unconventional secretory processing diversifies neuronal ion channel properties" for consideration by *eLife*. Your article has been reviewed by three peer reviewers, and the evaluation has been overseen by David Ginty as the Reviewing Editor and Eve Marder as the Senior Editor. Bruno Gold (peer reviewer) has agreed to reveal their identity decision has been reached after consultation between the reviewers. Based on these discussions and the individual reviews below, we regret to inform you that your work will not be considered further for publication in *eLife*.

Summary:

In this study, the authors characterize the types and extent of N-glycosylation of cell-surface membrane proteins in neurons. Unlike in other tissues or cells, neurons express substantial amounts of "core-glycosylated" ER-type modified proteins on their cell-surface. Neurons possess vastly greater amounts of plasma membrane vs. cell volume than other cells, and significant amounts of the cellular proteins are synthesized in the dendrites, structures containing extensive ER compartment but little to no Golgi membrane compartments. The Authors suggest that neurons may heavily utilize an unconventional protein secretion pathway in which membrane proteins are synthesized in the ER and then trafficked directly to the plasma membrane. Functionally, the Authors have shown that the life-time of a specific protein GluA2, is determined by the type of N-glycosylation and that neuronal activity can determine the extent of ER-type N-glycosylation on cell-surface. A strength of the work is the use of combinatorial purification steps to isolate core-glycosylated proteins from the cell-surface followed by mass-spec identification. The use of different glycosidase enzymes to reveal that specific proteins contain core, mature or mixed N-glycosylation modifications is also convincing.

This study addresses a fundamental aspect of neuronal and cell biology. The implications of this work are important and the findings are likely to be of interest to a broad audience. However, while some of the key findings are convincing, there are major concerns with both the cell biology and functional aspects of the study. The major concerns fall into two categories:

The evidence that dendritic proteins bypass the Golgi is weak. The physiological relevance of the observations is not well developed.

Given the extensive and deep concerns expressed by the reviewers, and the length of time it would take to generate additional data in support of your conclusions, we must decline to accept this work but encourage you to continue to pursue the problem. Please do consider a new submission to *eLife* if you are able to address the issues raised by the reviewers.

Major concerns:

1) Figure 1 and Figure 3 use specific lectins to show that neurons express large amounts of core glycoproteins on the cell surface and at synapses. Although presented as quantitative, it is not clear whether the flourescent labeling intensity (Figure 1) provides information about the relative abundance of cell surface glycans with different structures. Also, the only control for these experiments is glycosidase treatment of permeabilized COS7 cells. The signal observed for different lectins will depend upon the biotin conjugation density of the different lectins and degree of non-specific background binding. Notably, lectin binding decreased by less than 2-fold with N-glycanase treatment (Figure 1) suggesting that nonspecific binding is an issue. The specificity of lectin staining and of the pharmacological treatments should be further tested in experiments in which neurons are treated with tunicamycin (Tm), swainsonine (Sw), kifunensine (Kf), or deoxymannojirimycin (DMJ) and the staining intensity/patterns of surface lectin labeling is then assessed. The Authors could also compare the lectin staining patterns of surface and total (permeabilized) lectin labeling in neurons.

The far Western method used in Figure 2 shows results with better controls than surface lectin staining. Again, the specificity and effectiveness of the pharmacological treatments would be clearly demonstrated if the authors could show that Tm, Sw, Kf, DMJ treatment could alter the staining intensity/pattern in far Western. There was some debate amongst the reviewers about whether this experiment in Figure 2 is compelling. The Authors should describe more precisely how the experiments of Figure 2 were performed and interpreted/quantified.

A related issue is the assumption that ConA will only bind high mannose oligosaccharides that are exemplified by the generic "immature/core" structure shown in Figure 1. Hybrid structures, which can also have 1 or 2 terminal α-linked mannose residues also bind to ConA with high affinity. This problem impacts Figure 1, Figure 2 and the MS analysis in Figure 4. This issue needs to be discussed in the revised manuscript.

2) Figure 1 uses cultured hippocampal neurons at 40DIV, while Figure 3 uses neurons at 26DIV, and Figure 5 uses 11DIV. For all of the other experiments, especially for the mass-spec analysis, the age of the neurons used is not specified. The Methods section regarding neuron culture is inadequate and should be expanded, as should the figure legends to make it clear to the readers how the experiments were done. It seems highly likely that the pattern and degree of different N-glycosylation shows a developmental profile, in which case the use of neurons of vastly different (or unknown) ages may complicate the interpretation of these results. The authors should either choose a specific age of neuron for their experiments or test whether the patter of N-glycosylation does indeed show a developmental profile. It is curious that the authors decided to use neurons at 40DIV for Figure 1, was there a specific reason to use such old neurons. In many culture preparations, 40DIV may be at or beyond the limits of cell viability, in which case high levels of ConA surface staining may be a result of an artifact due to poor cell health. Some indication of the health of 40DIV neurons should be included in the revision.

Related to this, Figure 5 shows the use of pharmacological inhibitors to test how different N-glycosylation contributes to dendrite growth. The methods/legend describes that drug treatment began on 8DIV and cells were imaged on 11DIV, this time frame is on the late side to examine neurite growth. In the majority of studies examining neurite growth in culture, neurons younger neurons DIV1-7 are used to examine the peak of dendrite growth. Could the authors also include data on neurite growth of 8DIV old neurons prior to drug treatment? The examples given for Tm treated neurons look like 1-2DIV neurons with almost no dendrites, far fewer than what would be expected from 8DIV. This suggests that Tm treatment not only prevented neurite outgrowth but caused considerable retraction of dendrites, perhaps due to severe toxicity of Tm treatment. The fact that Kf, Sw, and DMJ did not greatly impair neurite growth (or cause retraction) supports the conclusions of the authors, however, as mentioned above there are no data to show that Kf, Sw, and DMJ are actually effective at limiting mature N-glycosylation. Lectin labeling and far-Western could be used to show the effectiveness of these drugs.

3) The Authors' conclusion that the plasma membrane proteins use a non-canonical secretory pathway is in part based upon the treatment of cells with brefeldin A (Figure 6). A limitation of this set of experiments is that the detection method (cell surface biotinylation) cannot distinguish between proteins that were transported to the cell surface before or after drug treatment. Cell surface expression will be determined by a combination of the externalization rate, internalization rate and degradation rate. If two proteins are compared that have different half-lives, the one with the shorter half-life will show the greatest reduction when cell surface expression is blocked by any treatment (BFA, cycloheximide, tunicamycin, etc.). The authors should address this critical issue.

4) A major issue raised by all three reviewers is that the physiological relevance of the observations is not adequately developed. In fact, the title of the paper implies that unconventional processing of membrane proteins "diversifies" neuronal ion channel properties. However, there are almost no data to support this.

Given that many surface receptors appear to be core-glycosylated, one would expect that treatment with ConA would have a strong effect on post-synaptic signaling. This could be easily tested. The effects of inhibitors of glycan maturation and glycosidase treatment should also be tested for their impact on Ca++ signaling or other aspects of neuronal physiology. These or related functional experiments should be done to increase the impact of the work.

Beyond this, the authors should limit their conclusions regarding "diversification" of neuronal ion channel properties. Alternatively, since the authors specifically address GluA2 and TARP processing, these proteins could be further examined in new experiments to strengthen this aspect of the work. Brefeldin A (BFA) or Kf treatment reduces the expression of surface TARPg8 but not (or less so) for GluA2. Do these treatments result in greater amount of TARPless AMPARs on the surface/ at synapses? The authors could examine the extent of GluA2/TARP interaction using co-immunoprecipitations, with the expectation that CoIP would be reduced following BFA or Kf treatment. Kf treatment did not show any effect on mEPSC amplitude or frequency. If significantly less TARP is expressed on the surface/synapses, then there may be important changes in AMPAR channel properties. The authors could also examine mEPSC decay kinetics or sensitivity of surface AMPAR to different drugs that modify channel properties in a TARP dependent manner. Such experiments are quite involved and likely beyond the scope of the present study, however, and thus simply toning down the conclusions/title seems the best way forward.

5) An experimental problem with the 2-stage enrichment MS procedure is readily observed when one examines the supplemental tables. Neither stage of the enrichment strategy is sufficiently robust to prevent false-positives. At best 66% of the proteins listed in Table III should be biotinylated in intact cells, and this value assumes that the secreted proteins were extracellular and remained bound to the cells when media was added prior to biotinylation. The false positives included proteins that are not glycoproteins (cytosol, nucleus mitochondria) as well as potential intracellular glycoproteins (ER, Golgi, vacuole) indicating that non-specific binding of proteins to ConA beads was also an issue. Given a 33% false positive rate, it is only reasonable to question whether a 33% false positive rate also applies to the plasma membrane and secreted proteins. The Authors should discuss this issue of false-positives in their ms dataset in the revised manuscript.

6) The experiment to test whether the different forms of GluA2 (Endo H sensitive or EndoH resistant) have different half lives assumes that the disappearance of the immature form of GluA2 from the plasma membrane is diagnostic of degradation. It has been known for more than 20 years that there are retrograde trafficking pathways from the cell surface back to the Golgi that allow additional Golgi glycan processing reactions on cell-surface glycoproteins. The authors are referred to multiple papers published by Martin Snider's laboratory between 1986 and 1996 including the following: (J.C.B. 103:265; J.B.C 264: 7675; Met Cell Biol. 32:339). Conversion of the immature GluA2 to the mature GluA2 is a reasonable alternative explanation for the more rapid loss of immature GluA2. This is an important point that the authors should discuss when describing the interpretation of results shown in Figure 7.

7) The implied model, not specifically described, is that dendritic proteins synthesized at distance from Golgi elements may be expressed on the cell-surface as core-glycosylated proteins, while proteins synthesized near Golgi elements or in the soma will be expressed with mature glycosylation. Previous studies have identified numbers of mRNAs that are trafficked to dendrites for "localized" protein synthesis. Is there any correlation between dendritic mRNA targeting and core-glycosylation? For example, is TARPg8 primarily synthesized in the soma whereas GluN1 or GABAARb3 are targeted to dendrites? This point should be discussed in the revised manuscript.

8) While perhaps not essential for publication, the manuscript would be greatly strengthened by some cell biological characterization of ER/Golgi or glycosyl transferases/glycosidases. For example, are core (ER) glycosyltransferases localized in dendrites or near synapses to a greater extent than Golgi-type modifying enzymes? This point should be discussed in the revision.

*Reviewer #1:*

In the current study by Hanus et al., the authors characterize the types and extent of N-glycosylation of cell-surface membrane proteins in neurons. Unlike in other tissues or heterologous cells, neurons express substantial amounts of "core-glycosylated" ER-type modified proteins on their cell-surface. Neurons possess vastly greater amounts of plasma membrane vs. cell volume than other cells, and significant amounts of the cellular proteins are synthesized in the dendrites, structures containing extensive ER compartment but little to no golgi membrane compartments. The implications of the current work are that neurons may heavily utilize an unconventional protein secretion pathway in which membrane proteins are synthesized in the ER and then trafficked directly to the plasma membrane. The authors have also shown that the life-time of a specific protein GluA2, is determined by the type of N-glycosylation and that neuronal activity can determine the extent of ER-type N-glycosylation on cell-surface. A major strength of the current work is the use of combinatorial purification steps to isolate core-glycosylated proteins from the cell-surface followed by mass-spec identification. The use of different glycosidase enzymes to reveal that specific proteins contain core, mature or mixed N-glycosylation modifications is also convincing.

This work addresses a fundamental aspect of neuronal and cell biology. The implications of this work are important and the findings are likely to be of interest to a broad audience. While some of the key findings of this work are convincing, the cell biology and functional aspects of the study are underdeveloped. The work would also be strengthened by several additional controls.

1) Figure 1 and Figure 3 use specific lectins to show that neurons express large amounts of core glycoproteins on the cell surface and at synapses. The only control for these experiments is glycosidase treatment of permeabilized COS7 cells. It seems highly probable that some of the surface lectin staining may be non-specific. The authors should also attempt glycosidase treatment to eliminate surface lectin labeling of neurons. In Figure 2, glycosidase treatment results in almost complete loss of signal using far-Western, and Figure 3 shows that glycosidase treatment results in a quantitative shift in electrophoretic mobility of specific proteins showing complete loss of glycosylation. However, the control experiment shown in 1D shows that glycosidase treatment reduces lectin binding by less than 50%, suggesting that the majority of signal from lectin staining is non-specific. This potential background staining may be even higher in neurons, hindering clear interpretation of Figure 1 and Figure 3. The specificity of lectin staining and of the pharmacological treatments would be further demonstrated if the authors could show that treatment of neurons with tunicamycin (Tm), swainsonine (Sw), kifunensine (Kf), or deoxymannojirimycin (DMJ) could alter the staining intensity/patterns of surface lectin labeling. It would also be interesting the compare the lectin staining pattern of surface and total (permeabilized) lectin labeling in neurons.

2) The far Western method used in Figure 2 shows more clear results with better controls than surface lectin staining. Again, the specificity and effectiveness of the pharmacological treatments would be clearly demonstrated if the authors could show that Tm, Sw, Kf, DMJ treatment could alter the staining intensity/pattern in far Western.

3) Figure 1 is described as using cultured hippocampal neurons at 40DIV, while Figure 3 uses neurons at 26DIV, and Figure 5 uses 11DIV. For all of the other experiments, especially for the mass-spec analysis, the age of the neurons used is not specified. The Methods section regarding neuron culture is inadequate and should be expanded, as well as the figure legends to make it clear to the readers how the experiments were done. It seems highly likely that the pattern and degree of different N-glycosylation may show a developmental profile, in which case the use of neurons of vastly different (or unknown) ages may complicate the interpretation of these results. The authors should either choose a specific age of neuron for their experiments or test whether the patter of N-glycosylation does indeed show a developmental profile. It is curious that the authors decided to use neurons at 40DIV for Figure 1, was there a specific reason to use such old neurons? In many culture preparations, 40DIV may be at or beyond the limits of cell viability, in which case high levels of ConA surface staining may be a result of an artifact due to poor cell health.

4) Related to the previous comment, Figure 5 shows the use of pharmacological inhibitors to test how different N-glycosylation contributes to dendrite growth. The methods/legend describes that drug treatment began on 8DIV and cells were imaged on 11DIV, this time frame is on the late side to examine neurite growth. In the majority of studies examining neurite growth in culture neurons younger neurons DIV1-7 are used to examine the peak of dendrite growth. Could the authors also include data on neurite growth of 8DIV old neurons prior to drug treatment? The examples given for Tm treated neurons look like 1-2DIV neurons with almost no dendrites, far fewer than what would be expected from 8DIV. This suggests that Tm treatment not only prevented neurite outgrowth but caused considerable retraction of dendrites, perhaps due to severe toxicity of Tm treatment. The fact that Kf, Sw, and DMJ did not greatly impair neurite growth (or cause retraction) supports the conclusions of the authors, however, as mentioned above there is no data to show that Kf, Sw, and DMJ are actually effective at limiting mature N-glycosylation. Lectin labeling and far-Western could be used to show the effectiveness of these drugs.

5) The title of the paper implies that unconventional processing of membrane proteins "diversifies" neuronal ion channel properties. However, there is almost no data to support this. The authors may wish to limit their conclusions on the functional consequences of core-glycosylation in favor of further characterization of unconventional secretory processing phenomenon, such as during development or by neuronal activity. Alternatively, since the authors specifically address GluA2 and TARPg8 processing these proteins could be further examined to strengthen the functional conclusions of the work. Brefeldin A (BFA) or Kf treatment reduces the expression of surface TARPg8 but not (or less so) for GluA2. Do these treatments result in greater amount of TARPless AMPARs on the surface/ at synapses? The authors could examine the extent of GluA2/TARP interaction using co-immunoprecipitation, with the expectation that CoIP would be reduced following BFA or Kf treatment. Kf treatment did not show any effect on mEPSC amplitude or frequency. If significantly less TARP is expressed on the surface/synapses there may be important changes in AMPAR channel properties. The authors could also examine mEPSC decay kinetics or sensitivity of surface AMPAR to different drugs that modify channel properties in a TARP dependent manner. Does acute loss of surface glycosylation by glycosidase treatment alter synaptic properties?

6) Figure 7 shows that core-glycosylated GluA2 has a shorter half-life than mature glycosylated GluA2. This is very interesting data that strengthens this work. However, this is the only data showing that unconventional secretory processing diversifies neuronal ion channel properties. The authors also show using their far-Western method that blocking excitatory synaptic transmission increases the surface abundance of core-glycosylation. Can this result be reproduced using surface lectin staining? Can the authors show that core-glycosylation of GluA2 is also increased by AP5/CNQX treatment? If this is the case than it would also be anticipated that the half-life of GluA2 would be reduced following AP5/CNQX treatment. However, this seems unlikely given that previous studies show that inactivity reduces AMPAR surface turnover (Ehlers, Neuron 2000) and increases AMPAR half-life (O'Brien et al., Neuron 1998). Currently the data on neuronal activity and glycosylation is underdeveloped. Either, Figure 7 could be removed, or the effect of neuronal activity on surface glycosylation could be further characterized.

7) The implied model, not specifically described, is that dendritic proteins synthesized at distance from golgi elements may be expressed on the cell-surface as core-glycosylated proteins, while proteins synthesized near golgi elements or in the soma will be expressed with mature glycosylation. Previous studies have identified numbers of mRNAs that are trafficked to dendrites for "localized" protein synthesis. Is there any correlation between dendritic mRNA targeting and core-glycosylation? For example, is TARPg8 primarily synthesized in the soma whereas GluN1 or GABAARb3 are targeted to dendrites? This point could be discussed. Finally, while perhaps not essential for publication, the manuscript would be greatly strengthened by some cell biological characterization of ER/golgi or glycosyl transferases/glycosidases. For example, are core (ER) glycosyltransferases localized in dendrites or near synapses to a greater extent than golgi-type modifying enzymes?

*Reviewer #2:*

The manuscript from Hanus et al. reports that neuronal cells either have a higher proportion of cell surface glycoproteins bearing high mannose oligosaccharides than several fibroblast cell lines. The authors would like to conclude that this difference in oligosaccharide structure is explained by trafficking of neuronal proteins to the plasma membrane by a "non-canonical secretory pathway." The authors would also like to conclude that the presence of immature oligosaccharide structures on a cell surface glycoprotein (GluA2) reduces the stability of the glutamate receptor as part of a regulatory mechanism. Unfortunately, there are a number of conceptual and technical concerns with this manuscript that preclude publication in *eLife*.

1) The authors appear to view Golgi processing events as obligatory events for typical cell-surface and secreted proteins. Retention of some high mannose oligosaccharides on N-linked glycoproteins is not that rare, nor is it restricted to proteins synthesized by neuronal cells. HIV gp120 retains a number of high-mannose glycans. ConA capture has been used as a method to capture glycoproteins for mass spectrometry in secreted fluids (serum and urine). Golgi processing of glycans is influenced by accessibility of the core structure to the Golgi mannosidases and glycosyltransferases.

2) A major issue in this manuscript is the assumption that ConA will only bind high mannose oligosaccharides that are exemplified by the generic "immature/core" structure shown in Figure 1. Hybrid structures, which can also have 1 or 2 terminal α-linked mannose residues also bind to ConA with high affinity. This problem with specificity impacts Figure 1, Figure 2 and the MS analysis in Figure 4.

3) Although presented as quantitative, it is not clear whether the flourescent labeling intensity (Figure 1) provides information about the relative abundance of cell surface glycans with different structures. The signal observed for different lectins will depend upon the biotin conjugation density of the different lectins and degree of non-specific background binding. Notably, lectin binding decreased by less than 2-fold with N-glycanase treatment (Figure 1) suggesting that nonspecific binding is an issue.

4) Banding patterns for surface ConA and RCA in neuronal cells seem to be different in panels B and C of Figure 2. Prominent bands that are both ConA reactive and RCA reactive are apparent in Figure 2. The ConA lanes are overloaded in Figure 2, while the RCA pattern differs in 2B and 2C show a different number of major bands (~5 vs. ~8). Internal samples in Figure 2 are apparently underloaded, as little signal is detected even though the final quantification (Figure 2) indicates that only ~35% of ConA reactive glycoproteins were present in the surface fraction of neuronal cells.

5) A conceptual problem with the 2-stage enrichment procedure used for the MS analysis is well illustrated by the examples of GluA1 and GluA2. GluA1 has six glycosylation acceptor sites in the extracellular domain. Based upon the molecular weight shift caused by EndoH digestion (Figure 3) the majority of these glycans are EndoH resistant. However, one high mannose oligosaccharide/protein is sufficient to allow ConA capture, and eventual inclusion into the "core glycosylated protein category". GluA2 is an example of a protein that was detected as a mixture of immature and mature forms. The minor form (not quantified, but clearly much less than 50%) is Endo H sensitive. This minor population is also sufficient to allow GluA2 to be categorized in the core glycosylated protein category (Figure 4 and supplemental tables) even though the majority of the protein has complex and/or hybrid oligosaccharides.

6) An experimental problem with the 2-stage enrichment MS procedure is readily observed when one examines the supplemental tables. Neither stage of the enrichment strategy is sufficiently robust to prevent false-positives. At best 66% of the proteins listed in Table III should be biotinylated in intact cells, and this value assumes that the secreted proteins were extracellular and remained bound to the cells when media was added prior to biotinylation. The false positives included proteins that are not glycoproteins (cytosol, nucleus mitochondria) as well as potential intracellular glycoproteins (ER, Golgi, vacuole) indicating that non-specific binding of proteins to ConA beads was also an issue. Given a 33% false positive rate, it is only reasonable to question whether a 33% false positive rate also applies to the plasma membrane and secreted proteins.

7) The author's conclusion that the plasma membrane proteins use a non-canonical secretory pathway is in part based upon the treatment of cells with brefeldin A (Figure 6). The limitation of this set of experiments is that the detection method (cell surface biotinylation) cannot distinguish between proteins that were transported to the cell surface before or after drug treatment. Cell surface expression will be determined a combination of the externalizaiton rate, internalization rate and degradation rate. If two proteins are compared that have different half-lives, the one with the shorter half-life will show the greatest reduction when cell surface expression is blocked by any treatment (BFA, cycloheximide, tunicamycin, etc.).

8) The experiment to test whether the different forms of GluA2 (Endo H sensitive or EndoH resistant) have different half lives assumes that the disappearance of the immature form of GluA2 from the plasma membrane is diagnostic of degradation. It has been known for more than 20 years that there are retrograde trafficking pathways from the cell surface back to the Golgi that allow additional Golgi glycan processing reactions on cell-surface glycoproteins. I refer the authors to multiple papers published by Martin Snider's laboratory between 1986 and 1996 including the following: (J.C.B. 103:265; J.B.C 264: 7675; Met Cell Biol. 32:339). Conversion of the immature GluA2 to the mature GluA2 is a reasonable alternative explanation for the more rapid loss of immature GluA2.

9) To obtain any evidence that the neuronal plasma membrane proteins are transported to the cell membrane by a "non-conventional secretory pathway" or by passage through a Golgi that is deficient in mannosidase activities it would be important to express a panel (4 or 5) of the candidate neuronal glycoproteins in a heterologous expression system to determine whether the glycosylation patterns differ. Conceptually, the observation that some neuronal proteins have mature oligosaccahrides, mixtures of mature and immature oligosaccharides, or mainly immature oligosaccharides) is inconsistent with the "non-conventional secretory pathway", as this would necessitate sorting events that allow Golgi skipping for subsets of proteins.

*Reviewer #3:*

In this manuscript, Hanus and colleagues report that an unexpected high number of surface proteins (hundreds), including neurotransmitter receptors, voltage-dependent ion channels, and neurotransmitter transporters are core-glycosylated (i.e. contain the core-glycan added to nascent proteins in the ER but not trimmed and processed by ER/Golgi glycosidases and glycosyltransferases) in hippocampal neurons in culture. This result was obtained by surface biotinylation, pulse-chase labeling, affinity purification and mass spectrometry experiments. Although a complete block of N-glycosylation using tunicamycin impairs dentritic growth, inhibitors of core-glycan maturation have no effect, suggesting that mature N-glycans are not required for dendritic development. The authors provide evidence that core-glycosylated proteins reach the cell surface by Golgi by-pass pathways. Finally, by studying the turn-over of the AMPA receptor subunit GluA2 that exists at the plasma membrane under mature and immature forms, they document that the core-glycosylated pool of GluA2 has a substantially shorter half-time than the pool with mature N-glycans.

This is an interesting piece of work that could have important physiological consequences. It also explains the blocking effect on the desensitization of glutamate and kainate receptors of concanavalin A (ConA), a lectin that binds to core-glycans and is classically used by electrophysiologists. The experiments are well done and generally convincing.

Specific comments:

1) Given that many surface receptors appear to be core-glycosylated, one could expect that the treatment with ConA will have a strong effect on post-synaptic signaling. This could be easily test (for instance by measuring Ca++ activity) and could improve the impact of this study.

2) Figure 1: in panel D, the authors confirm the specificity of lectin labeling using permeabilized COS-7 cells. On the other hand, in panel C, the level of core-glycosylated N-glycans is measured in non-permeabilized neuronal cells. As the treatment with EndoH and PNGase does not fully inhibit ConA or WGA signal (about 50%), it seems important to treat non-permeabilized neurons with EndoH and PNGase and to measure lectin binding.

3) Figure 6: the experiments with BFA are not fully convincing. In panel B, BFA treatment seems to decrease the surface expression of the three receptors, including GLuN1. This experiment should be quantified. There are two lanes for each condition (Ct and BFA): do they correspond to duplicates? Also, the treatment with BFA seems pretty long (6-7h) whereas a dispersion of Golgi is already observed after 2h (panel A). What is the reason? Could it affect cell viability? Finally, it is unclear to me how the results shown in panel C have been obtained.

---

## [Author Response]

*The manuscript has been improved but there are a few remaining issues that need to be addressed before acceptance, as outlined below:*

*The reviews wonder why the blots of BONCAT samples in Figure 6 and Figure 6—figure supplement 1 look so different. Particularly the left panel in Figure 6 appears unusual – the strongest signal is at the edges of the lane.*

The different labeling pulse durations (~3 vs. 5 hours) and resulting low or higher signal intensities in Figure 6 and Figure 6—figure supplement 1 may give the impression that the two blots are qualitatively different. Yet, the unusual migration pattern that is particularly obvious for the surface samples in Figure 6 was repeatedly observed in these experiments.

We have now replaced the blots in Figure 6—figure supplement 1 to show a broader set of conditions and biological replicates, to show the repeated occurrence of this unusual migration pattern and to further document the lack of effect of BFA on secretory trafficking (here in contrast to Figure 6, without pre-incubation prior to metabolic labeling.)

The reason for this pattern is not totally clear. We suspect that it is due both to the exposure of surface samples to high levels of detergents during their elution after purification (~0.6% SDS, ~1% Triton and 50mM DTT) and the further precipitation and salt exchange of the samples (both surface and total) that was needed to eliminate DTT, as, in our hands, this compound inhibits CLICK reactions.

*There is also a question about the combination of surface biotinylation and BONCAT. For the experiment shown in Figure 6, surface proteins were biotinlylated for the separation of surface and intracellular proteins, it is assumed by affinity purification, then again biotinylated in the BONCAT procedure, and then separated by SDS-PAGE and immunodetected by Western blotting with anti-biotin antibodies. This means, that the proteins analyzed in the surface protein fraction are biotinylated irrespective of the BONCAT treatment. Does this not confound the analysis and the interpretation that nascent proteins are analyzed specifically? Please address this issue in the resubmission.*

This is not correct. Surface proteins were labeled with NHS-SS-biotin which can be cleaved by reduction, enabling an efficient and straightforward means to elute surface proteins from streptavidin beads after purification. This reduction results in the complete removal of biotin from surface proteins as demonstrated by the lack of further interaction with streptavidin beads (see lane “0 uL ConA” in Figure 4) or lack of detection by anti-biotin antibodies (see lanes “Met” in Figure 6—figure supplement 1). Nevertheless, we thank the reviewer(s) for pointing this out and have modified the manuscript to clarify this issue (see subsection “Surface-biotinylation and pulse chase experiments”).

[Editors’ note: the author responses to the first round of peer review follow.]

*Major concerns:*

*1) Figure 1 and Figure 3 use specific lectins to show that neurons express large amounts of core glycoproteins on the cell surface and at synapses. Although presented as quantitative, it is not clear whether the flourescent labeling intensity (Figure 1) provides information about the relative abundance of cell surface glycans with different structures. Also, the only control for these experiments is glycosidase treatment of permeabilized COS7 cells. The signal observed for different lectins will depend upon the biotin conjugation density of the different lectins and degree of non-specific background binding. Notably, lectin binding decreased by less than 2-fold with N-glycanase treatment (Figure 1) suggesting that nonspecific binding is an issue. The specificity of lectin staining and of the pharmacological treatments should be further tested in experiments in which neurons are treated with tunicamycin (Tm), swainsonine (Sw), kifunensine (Kf), or deoxymannojirimycin (DMJ) and the staining intensity/patterns of surface lectin labeling is then assessed. The Authors could also compare the lectin staining patterns of surface and total (permeabilized) lectin labeling in neurons.*

The point of the reviewer is well taken. In contrast to immunoblotting and far western blotting where these glycosydases completely abrogate glycoprotein recognition by lectins (see Figure 2), EndoH and PNGase have only a partial effect when used for cell labeling and imaging experiments. This discrepancy strongly suggests that these enzymes (EndoH and PNGase) are only fully efficient when used with denatured, solubilized proteins. Although we tried these deglycosylation experiments for imaging using various permeabilization conditions, we did not detect stronger effects. We note, however, that this partial effect and resulting high background signal could also indicate that we may greatly underestimate the levels of surface core-glycosylated proteins in neurons versus other cell types. We also note that to avoid biases in the quantification, we did not threshold fluorescence and quantified the mean fluorescence across the entire cell, which partly contributed to high background signals.

Based on these observations, we relied primarily on far western blotting to compare the surface levels of these glycoproteins and, as suggested by the reviewer, used this assay to test a cast of glycosylation inhibitors (see response to the next point below). Nevertheless, to address reviewer’s comment, we compared the surface levels of ConA-reactive glycans in control neurons and in neurons treated with kifunensine and confirmed that this inhibitor increases the levels of surface labeling with ConA and GNA while decreasing the levels of surface RCA and WGA (Figure 1—figure supplement 2).

We now include imaging data in permeabilizing conditions and document the localization of intracellular core, hybrid, and complex N-glyans in distinct organelles (Figure 1—figure supplement 3).

We now also include additional imaging data in COS7 cells where secretory organelles are easily identifiable by their subcellular localization and their morphology. We now document a differential effect of PNGase and EndoH on specific secretory compartments. EndoH selectively abolishes the signal associated with ConA-reactive immature glycoproteins in the endoplasmic reticulum, but has no effect on the levels of RCA or WGA-reactive species in the Golgi. In contrast, PNGase strongly decreases the labeling of the Golgi by RCA or WGA or the ER by ConA (not shown) (revised Figure 1—figure supplement 4).

*The far Western method used in Figure 2 shows results with better controls than surface lectin staining. Again, the specificity and effectiveness of the pharmacological treatments would be clearly demonstrated if the authors could show that Tm, Sw, Kf, DMJ treatment could alter the staining intensity/pattern in far Western.*

We thank the reviewer for her/his suggestion. We performed these experiments for Kf using surface staining and imaging (Figure 1—figure supplement 1

2) and, for Kf, DMJ and Sw surface biotinylation and far western blotting (Figure 5—figure supplement 1). As expected, Tm decreases the surface levels of ConA, RCA and WGA reactive proteins. Kf, Sw and DMJ strongly increase the levels of ConA reactive species while markedly decreasing the levels of RCA reactive species. Kf and DMJ decrease the levels of WGA species, but these are unexpectedly increased by Sw. Although the reason for this later observation is not clear at present, these experiments clearly confirm the specificity of our quantification method and, importantly, validate our conclusions regarding the effect of these drugs on dendritic development and maintenance (see also our last response to point 1 below).

*There was some debate amongst the reviewers about whether this experiment in Figure 2 is compelling. The Authors should describe more precisely how the experiments of Figure 2 were performed and interpreted/quantified.*

We modified the manuscript to provide more details on these experiments and how they were quantified. See subsection “Quantification of protein surface expression after surface biotinylation” and Figure legend 2.

*A related issue is the assumption that ConA will only bind high mannose oligosaccharides that are exemplified by the generic "immature/core" structure shown in Figure 1. Hybrid structures, which can also have 1 or 2 terminal α-linked mannose residues also bind to ConA with high affinity. This problem impacts Figure 1, Figure 2 and the MS analysis in Figure 4. This issue needs to be discussed in the revised manuscript.*

The reviewer raises a valid concern. The revised manuscript has been modified accordingly (see subsection “Distinction of core and hybrid N-glycans”). We note however that our data show that we can indeed distinguish core-glycosylated from hybrid N- glycans:

As exemplified in Figure 1 and Figure 5, hybrid N-glycans are typically recognized by RCA. But, as clearly shown in Figure —figure supplement 2, RCA-binding proteins are EndoH *insensitive* in hippocampal neurons, in contrast to ConA binding proteins which are EndoH sensitive;

And, as clearly shown in Figure 1—figure supplement 2 and Figure 5—figure supplement 1, RCA-reactive proteins respond completely differently from ConA-reactive species to Kf, DMJ and Sw.

Thus, although we cannot exclude that rare Endo-H sensitive hybrid N- glycans that bind ConA but not RCA exist in neurons, our data strongly indicate that core-glycans are atypically abundant at the neuronal surface.

*2) Figure 1 uses cultured hippocampal neurons at 40DIV, while Figure 3 uses neurons at 26DIV, and Figure 5 uses 11DIV. For all of the other experiments, especially for the mass-spec analysis, the age of the neurons used is not specified. The Methods section regarding neuron culture is inadequate and should be expanded, as should the figure legends to make it clear to the readers how the experiments were done. It seems highly likely that the pattern and degree of different N-glycosylation shows a developmental profile, in which case the use of neurons of vastly different (or unknown) ages may complicate the interpretation of these results. The authors should either choose a specific age of neuron for their experiments or test whether the patter of N-glycosylation does indeed show a developmental profile.*

Most of the experiments were done in mature cultured neurons, well beyond the stage at which synapses are forming or are in flux.

Nevertheless, we wanted to directly address this point. To address this, we monitored the surface levels of ConA and RCA reactive glycans over the course of 9 weeks and detected equally high levels of ConA reactive species throughout neuronal development and maturation. These experiments thus justify the validity of using neurons at distinct ages. Importantly, we found that in contrast to ConA-reactive glycans, the surface expression of RCA-reactive- glycans progressively increases with neuronal maturation, documenting a developmental regulation of the glycosylation profile of neuronal surface proteins. See revised Figure 2.

*It is curious that the authors decided to use neurons at 40DIV for Figure 1, was there a specific reason to use such old neurons. In many culture preparations, 40DIV may be at or beyond the limits of cell viability, in which case high levels of ConA surface staining may be a result of an artifact due to poor cell health. Some indication of the health of 40DIV neurons should be included in the revision.*

We respectfully disagree with the reviewer and take pride in being able to consistently maintain cultured neurons in good health over long periods of time. As mentioned above, we can now rule out that the high expression of surface ConA reactive glycans in neurons is an artifact due to neuron “excessive” age.

As shown in revised Figure 1, Figure 1—figure supplement 1, Figure 2, Figure 2—figure supplement 1 and Figure 3, the neurons that were used for these experiments showed no sign of poor health as indicated by the surface expression of neurotransmitter receptors, the morphology and density of their synapses, their MAP2 staining and the integrity of their plasma membrane.

*Related to this, Figure 5 shows the use of pharmacological inhibitors to test how different N-glycosylation contributes to dendrite growth. The methods/legend describes that drug treatment began on 8DIV and cells were imaged on 11DIV, this time frame is on the late side to examine neurite growth. In the majority of studies examining neurite growth in culture, neurons younger neurons DIV1-7 are used to examine the peak of dendrite growth. Could the authors also include data on neurite growth of 8DIV old neurons prior to drug treatment? The examples given for Tm treated neurons look like 1-2DIV neurons with almost no dendrites, far fewer than what would be expected from 8DIV. This suggests that Tm treatment not only prevented neurite outgrowth but caused considerable retraction of dendrites, perhaps due to severe toxicity of Tm treatment.*

We agree with the reviewer about the fact that Tm induces a retraction of dendrites and modified the text to emphasize that Tm prevents dendritic growth and maintenance(see subsection “Mature N-glycans are not required for dendritic development and maintenance”).

We however respectfully disagree with the reviewer about the non-specific toxicity of Tm. Although the exact turnover rate of dendritic proteins is still a matter of debate, their average stability is in the order of 2-3 days (Cohen et al., 2013). Because N-glycosylation is necessary for the processing of a large number of dendritic membrane proteins, it is expected that blocking the renewal of these proteins over 3-4 days will induce dendritic shrinkage.

*The fact that Kf, Sw, and DMJ did not greatly impair neurite growth (or cause retraction) supports the conclusions of the authors, however, as mentioned above there are no data to show that Kf, Sw, and DMJ are actually effective at limiting mature N-glycosylation. Lectin labeling and far-Western could be used to show the effectiveness of these drugs.*

We fully agree with the reviewers and did these experiments, which, as mentioned above (see our third response to point 1), confirmed that the drugs had the expected effects on neuronal surface N-glycans. These new experiments strongly support our conclusion that core-glycosylated proteins are indeed sufficient to sustain dendritic growth. See revised Figure 1—figure supplement 2 and Figure 5—figure supplement 1.

*3) The Authors' conclusion that the plasma membrane proteins use a non-canonical secretory pathway is in part based upon the treatment of cells with brefeldin A (Figure 6). A limitation of this set of experiments is that the detection method (cell surface biotinylation) cannot distinguish between proteins that were transported to the cell surface before or after drug treatment. Cell surface expression will be determined by a combination of the externalization rate, internalization rate and degradation rate. If two proteins are compared that have different half-lives, the one with the shorter half-life will show the greatest reduction when cell surface expression is blocked by any treatment (BFA, cycloheximide, tunicamycin, etc.). The authors should address this critical issue.*

The point of the reviewer is well-taken and we agree that this is a key issue. To address this, we selectively monitored the effect of BFA on the accumulation of nascent proteins at the surface after protein metabolic labeling with a bio-orthogonal strategy with a non-canonical amino acid (azido-homoalanine). As shown in revised Figure 8, BFA completely abolishes the delivery of nascent proteins to the cell surface in COS cells. In stark contrast and as we expected, BFA has only a minor effect on the accumulation of these proteins in neurons, thus showing that unconventional secretory processing is atypically common in these cells.

In addition, we would like to stress that, in hippocampal neurons, GluN1 has a shorter half-life than GluA2 (1.61 vs. 1.95 days, respectively) (Cohen et al., 2013; reviewed in Hanus and Schuman, 2013). It is thus very unlikely that the distinct effects of BFA that we observed for these proteins is merely due to their having distinct stabilities. The stability of TARP γ 8 is unknown at present. This point is now discussed in the manuscript (see subsection “Core-glycosylated proteins traffic to the neuronal surface through a non-canonical secretory pathway”, second paragraph).

*4) A major issue raised by all three reviewers is that the physiological relevance of the observations is not adequately developed. In fact, the title of the paper implies that unconventional processing of membrane proteins "diversifies" neuronal ion channel properties. However, there are almost no data to support this.*

*Given that many surface receptors appear to be core-glycosylated, one would expect that treatment with ConA would have a strong effect on post-synaptic signaling. This could be easily tested. The effects of inhibitors of glycan maturation and glycosidase treatment should also be tested for their impact on Ca++ signaling or other aspects of neuronal physiology. These or related functional experiments should be done to increase the impact of the work.*

*Beyond this, the authors should limit their conclusions regarding "diversification" of neuronal ion channel properties.*

The reviewers raise an important point. However, we would like to stress that the fact that dendritic development and maintenance require N- glycosylation but does not rely on Golgi-dependent N-glycosylation is a novel functional finding and has important implications for our understanding of neuronal development and synaptic function.

Nevertheless, we performed additional experiments to address the issue. We decided to focus on the synaptic responses driven by AMPA receptors because of the co-existence of surface AMPA receptors with mature and core-glycosylated glycosylation profiles that we detected and reported in the manuscript. We thus sought to determine whether converting these proteins into mostly core-glycosylated profiles was sufficient to modulate neuron responses to glutamate. As described in revised Figure 8, we used calcium imaging and glutamate uncaging to selectively stimulate AMPA (/kainate) receptors in control conditions or after a 48h-incubation with kifunensine (which blocks the maturation of core-glycosylated N-glycans). We found that Kf regulates the kinetics of postsynaptic responses (decreased time to peak upon repeated stimulation) thus showing that core-glycosylation does modulate postsynaptic responses. These new data thus directly support the notion that unconventional secretory processing diversifies the properties of neuronal ion channels. See Revised Figure 7 and Figure 7—figure supplement 2.

*Alternatively, since the authors specifically address GluA2 and TARP processing, these proteins could be further examined in new experiments to strengthen this aspect of the work. Brefeldin A (BFA) or Kf treatment reduces the expression of surface TARPg8 but not (or less so) for GluA2. Do these treatments result in greater amount of TARPless AMPARs on the surface/ at synapses? The authors could examine the extent of GluA2/TARP interaction using co-immunoprecipitations, with the expectation that CoIP would be reduced following BFA or Kf treatment. Kf treatment did not show any effect on mEPSC amplitude or frequency. If significantly less TARP is expressed on the surface/synapses, then there may be important changes in AMPAR channel properties. The authors could also examine mEPSC decay kinetics or sensitivity of surface AMPAR to different drugs that modify channel properties in a TARP dependent manner. Such experiments are quite involved and likely beyond the scope of the present study, however, and thus simply toning down the conclusions/title seems the best way forward.*

We fully agree with the reviewer. We are interested in addressing this issue in more detail, which, as the reviewer noted, is not a trivial undertaking. We thus decided to remove the data that pertained to this from the present study as these experiments only hinted at a modulation of receptors properties without directly addressing this possibility.

*5) An experimental problem with the 2-stage enrichment MS procedure is readily observed when one examines the supplemental tables. Neither stage of the enrichment strategy is sufficiently robust to prevent false-positives. At best 66% of the proteins listed in Table III should be biotinylated in intact cells, and this value assumes that the secreted proteins were extracellular and remained bound to the cells when media was added prior to biotinylation. The false positives included proteins that are not glycoproteins (cytosol, nucleus mitochondria) as well as potential intracellular glycoproteins (ER, Golgi, vacuole) indicating that non-specific binding of proteins to ConA beads was also an issue. Given a 33% false positive rate, it is only reasonable to question whether a 33% false positive rate also applies to the plasma membrane and secreted proteins. The Authors should discuss this issue of false-positives in their ms dataset in the revised manuscript.*

We modified the manuscript to include a comment about possible false positives (see subsection “Hundreds of key surface neuronal membrane proteins are core-glycosylated”, third paragraph). We do however respectfully disagree with what the reviewer seems to imply about the specificity of our purification procedure:

Revised Figure 2 clearly shows our ability to selectively capture surface membrane proteins with our surface-biotinylation assay;

We included additional controls in revised Figure 2—figure supplement 1 that further show that the integrity of the plasma membrane is not compromised by this protocol;

We note that our sequential purification strategy is not properly speaking “denaturating” (SDS concentration during binding is 0.2 to 0.05%) and it is thus expected to find some of the many cytoplasmic partners of transmembrane surface proteins;

Despite this and as shown in revised Figure 4, these two purification steps enable us to selectively purify AMPA receptors from TARP gamma8, while these proteins are typically found together in immuno-precipitation experiments (Schwenk et al., 2014);

Figure 4—figure supplement 2 clearly shows the striking enrichment of N- glycosylated proteins in our dataset;

Finally, we would like to stress that the localization terms (ER, Golgi, vacuole) that are provided in the Tables are based on annotations (mostly from studies in non-neuronal cells) and should not be taken at face value. That being said, we note that the over-representation of ER proteins vs Golgi proteins is completely consistent with our view that Golgi bypass may be atypically frequent in neurons.

*6) The experiment to test whether the different forms of GluA2 (Endo H sensitive or EndoH resistant) have different half lives assumes that the disappearance of the immature form of GluA2 from the plasma membrane is diagnostic of degradation. It has been known for more than 20 years that there are retrograde trafficking pathways from the cell surface back to the Golgi that allow additional Golgi glycan processing reactions on cell-surface glycoproteins. The authors are referred to multiple papers published by Martin Snider's laboratory between 1986 and 1996 including the following: (J.C.B. 103:265; J.B.C 264: 7675; Met Cell Biol. 32:339). Conversion of the immature GluA2 to the mature GluA2 is a reasonable alternative explanation for the more rapid loss of immature GluA2. This is an important point that the authors should discuss when describing the interpretation of results shown in Figure 7.*

We thank the reviewer for bringing our attention to this point. We modified the manuscript, which now mentions this alternative possibility (see Discussion, seventh paragraph) which we nevertheless still find unlikely, given the paucity of golgi membrane in the dendrites.

*7) The implied model, not specifically described, is that dendritic proteins synthesized at distance from Golgi elements may be expressed on the cell-surface as core-glycosylated proteins, while proteins synthesized near Golgi elements or in the soma will be expressed with mature glycosylation. Previous studies have identified numbers of mRNAs that are trafficked to dendrites for "localized" protein synthesis. Is there any correlation between dendritic mRNA targeting and core-glycosylation? For example, is TARPg8 primarily synthesized in the soma whereas GluN1 or GABAARb3 are targeted to dendrites? This point should be discussed in the revised manuscript.*

We agree with the reviewer that these aspects are important. Arguably however, delineating where receptors are synthesized, folded, assembled, glycosylated and what exact itinerary they follow to reach the plasma membrane is largely unknown, and that these questions extend far beyond the scope of a single paper. In fact, this question constitutes the major project of another postdoctoral fellow in the lab. We hope, given the scope of what is covered in the revised manuscript, that the reviewers and editors agree that the including findings are of fundamental importance and indeed motivate future studies like those suggested above.

*8) While perhaps not essential for publication, the manuscript would be greatly strengthened by some cell biological characterization of ER/Golgi or glycosyl transferases/glycosidases. For example, are core (ER) glycosyltransferases localized in dendrites or near synapses to a greater extent than Golgi-type modifying enzymes? This point should be discussed in the revision.*

We modified the manuscript according to reviewer’s suggestion and now specifically discuss this point. A difficulty here is that as far as we can discern, the commercially available antibodies directed against these proteins do not work in rodent neurons. All the antibodies that we have tried so far did not work, yielding either no signal at all or unspecific signal (e.g. strong nucleus staining etc.). We note however that the exclusion of Golgi-GFP (derived from a human galactosyl-transferase) and endogenous Golgi Mannosidase II (with an antibody that is no longer available) from dendrites have been extensively documented (reviewed in (Hanus and Ehlers, 2008; 2016). (See also our second response to point 1).